# Aerosol solar radiative forcing near the Taklimakan Desert based on radiative transfer and regional meteorological simulations during the Dust Aerosol Observation-Kashi campaign

Li Li[1], Zhengqiang Li[1], Wenyuan Chang[2], Yang Ou[1], Philippe Goloub[3], Chengzhe Li[4], Kaitao Li[1], Qiaoyun Hu[3], Jianping Wang[1], Manfred Wendisch[5]

[1]Aerospace Information Research Institute, Chinese Academy of Sciences, Beijing 100101, China
[2] Institute of Atmospheric Physics, Chinese Academy of Sciences, Beijing 100029, China
[3] Laboratoire d'Optique Atmosphérique, Université de Lille 1/CNRS, Lille 59655, France
[4] Department of Chemical and Biochemical Engineering, University of Iowa, Iowa 52242, USA
[5] Leipzig Institute for Meteorology, Leipzig University, Leipzig 04103, Germany

*Correspondence to*: Zhengqiang Li (lizq@aircas.ac.cn)

**Abstract.** The Taklimakan Desert is a main and continuous source of Asian dust particles causing significant direct radiative effects, which are commonly quantified by the aerosol solar radiative forcing (*ASRF*). To improve the accuracy of estimates of dust *ASRF*, the Dust Aerosol Observation-Kashi (DAO-K) campaign was carried out near the Taklimakan Desert in April 2019. The objective of the DAO-K campaign is to provide crucial parameters needed for the calculation of *ASRF*, such as dust optical and microphysical properties, vertical distribution,  and surface albedo. The *ASRF* was calculated using radiative transfer (RT) simulations based on the observed aerosol parameters, additionally considering the measured atmospheric profiles and diurnal variations of surface albedo. As a result, daily average values of *ASRF* of -19 W m$^{-2}$ at the top of atmosphere and -36 W m$^{-2}$ at the bottom of atmosphere were derived from the simulations conducted during the DAO-K campaign. Furthermore, the Weather Research and Forecasting model with Chemistry (WRF-Chem), with assimilations of measurements of the aerosol optical depth and particulate matter (*PM*) mass concentrations of particles with aerodynamic diameter smaller than 2.5 μm (*PM*$_{2.5}$) and 10 μm (*PM*$_{10}$), is employed to estimate the dust *ASRF* for comparison. The results of the *ASRF* simulations (RT and WRF-Chem) were evaluated using ground-based observations of downward solar irradiances, which have confirmed that the RT simulations are in good agreement with simultaneous observations, whereas the WRF-Chem estimations reveal obvious discrepancies with the solar irradiance measurements.

## 1 Introduction

Atmospheric aerosol particles play a vital role in regional and global climate changes, directly by modifying the radiative balance of the Earth-atmosphere system, and indirectly by altering cloud radiative properties, as well as cloud development and precipitation through acting as cloud condensation nuclei (CCN) and/or ice nucleating particles (INP) (Twomey, 1977; IPCC, 2007; Lenoble et al., 2013; Werner et al., 2014). Mineral dust is the most abundant large aerosol type in the atmosphere

(Ansmann et al., 2011), which has a tremendous impact on the radiation budget, not only through scattering process, but also due to absorption of solar (0.3~5 µm), also called shortwave (SW) radiation (Otto et al., 2007; García et al., 2012; Valenzuela et al., 2012; Lenoble et al., 2013), with potential dynamic consequences (Wendisch et al., 2008; Li et al., 2017). Atmospheric dust particles may also alter the cloud properties by serving as CCN, giant CCN, and INP (Yin et al., 2002; DeMott et al.,

2003; van den Heever et al., 2006). Numerous efforts have been undertaken to investigate the solar radiative effects of mineral dust using radiative transfer (RT) models (e.g., Santa Barbara  DISORT Atmospheric Radiative Transfer (SBDART), Fu-Liou RT model), or  regional and global meteorological and climate models (e.g., Weather Research and Forecasting model with Chemistry (WRF-Chem), Regional Climate Model version 4 (RegCM4))  employing in-situ and remote sensing observations into the simulations (Huang et al., 2009, 2014; Sun et al., 2012; Chen et al., 2013, 2014, 2018; Li et al., 2018). However, the

quantification of the dust radiative effects is still challenging due to the high aerosol variability in space and time and the complex light scattering properties of mineral dust. Moreover, the dust radiative effects depend on the surface albedo over the desert and the cloud layer in the vertical as well (Bierwirth et al., 2009; Waquet et al., 2013; Xu et al., 2017).

   As one of the largest sandy deserts in the world, the Taklimakan Desert located in the Xinjiang Uygur Autonomous Region of China is a main source region of Asian dust (Huang et al., 2009). It influences not only surrounding areas such as

the Tibetan Plateau (Liu et al., 2008; Chen et al., 2013; Yuan et al., 2019), but also wide regions in Eastern Asia (Mikami et al., 2006; Liu et al., 2011a; Yuan et al., 2019), and even North America and Greenland through long-range transports across the Pacific Ocean (Bory et al., 2003; Chen et al., 2017; Liu et al., 2019). Therefore, an accurate assessment of the Taklimakan aerosol solar radiative forcing (*ASRF*, defined as the difference of the net solar irradiance with and without aerosols presence) is important to evaluate regional and global climate changes. However, the results of corresponding simulations of *ASRF*

applying different models with variable observation inputs vary widely in the open literature. Huang et al. (2009) employed the Fu-Liou RT model to simulate the Taklimakan *ASRF* during the dust episodes in the summer of 2006, and reported that the dust particles result in average daily mean solar warming effect of 14 W m$^{-2}$ at the top of atmosphere (TOA), atmospheric warming effect of 79 W m$^{-2}$, and a surface cooling effect of -65 W m$^{-2}$. Sun et al. (2012) adopted the RegCM4 simulations and reported both negative values of the *ASRF* (i.e., cooling effects) of dust particles at the TOA and bottom of atmosphere (BOA)

with the strongest values (up to -4 W m$^{-2}$ and -25 W m$^{-2}$, respectively) in spring between 2000 and 2009 in the Taklimakan Desert region. Li et al. (2018) reported negative multi-year average values of the aerosol solar radiative forcing of -16 W m$^{-2}$ at the TOA and -18 W m$^{-2}$ at the BOA at the edge of the Taklimakan Desert, Kashi station based on the SBDART simulations. The simulated results of dust aerosol radiative forcing have rarely been confirmed, especially in the Taklimakan Desert (Xia et al., 2009). Occasionally the performance of various model-based *ASRF* estimates were evaluated against the observations

of aerosol optical depth (*AOD*), aerosol extinction profile, single scattering albedo (*SSA*), and particle size distribution (Zhao et al., 2010; Chen et al., 2014). Nevertheless, comparison of irradiance is indispensable to provide direct evidence for corroborating the *ASRF* simulated results.

   The knowledge of the optical, physical, chemical, and radiative properties of dust aerosol particles are crucial to derive the *ASRF* of dust particles. To precisely measure these important dust properties over the Taklimakan Desert, an intensive field

campaign named Dust Aerosol Observation-Kashi (DAO-K) was performed. One of the goals of theDAO-K field campaign is to provide high quality dataset on aerosol in this region to obtain accurate assessment of the Taklimakan *ASRF*. In this paper, we focus on estimating direct *ASRF* of the dust-dominated aerosol population using SBDART simulations with appropriate ground-based and satellite measurements of aerosol parameters, surface albedo, and atmospheric vertical profiles. The *ASRF* simulations are comprehensively evaluated by comparison with the results of WRF-Chem simulations, ground-based

irradiance measurements, as well as the AErosol RObotic NETwork (AERONET, http://aeronet.gsfc.nasa.gov) operational products (Holben et al., 1998).

      Sect. 2 includes a brief introduction of the DAO-K field campaign, and an overview of the multi-source observations and data. Methods for estimating *ASRF* by improving the inputs of atmospheric profiles and land surface albedo in the RT simulation, and by employing data assimilations in the WRF-Chem simulation, are described in Sect. 3. Sect. 4 presents the

results of *ASRF* simulated by the RT model during the field campaign and for some specific cases. The influences of the atmosphere and surface conditions on the results are discussed. The difference from the corresponding AERONET operational products are also analysed in this section. The comparison between the RT and WRF-Chem model simulations is discussed in Sect.5. Both the model simulations are evaluated based on the simultaneous irradiance measurements. Summary and conclusions are given in Sect. 6.

**2 Dust Aerosol Observation-Kashi field campaign**

**2.1 Observation site**

The Dust Aerosol Observation-Kashi (DAO-K) field campaign with comprehensive observations of physical, chemical, and optical properties of aerosol particles, solar radiation, vertical structures of the atmosphere, and land surface albedo in the Taklimakan Desert region was designed to provide high quality data for aerosol radiative forcing estimates. Kashi is located

at the edge of the Taklimakan Desert; it is surrounded by the Tianshan Mountains in the North, the Pamir Plateau in the West, and the Kunlun Mountains in the South (Fig. 1). The DAO-K field campaign was conducted at the Kashi campus of the Aerospace Information Research Institute, Chinese Academy of Sciences (39.50˚N, 75.93˚E, 1320 m above mean sea level). The campus hosts a long-term observation station within the Sun-sky radiometer Observation NETwork (SONET, www.sonet.ac.cn) (Li et al., 2018). In addition to the Kashi station near the Taklimakan Desert, SONET also maintains two

dust aerosol observation stations (i.e., Zhangye and Minqin stations) in the Gobi Desert which is another important source of Asian dust. Although some studies reported that the dust generated in Taklimakan Desert exerts a less influence on long-range downstream regions due to the unique terrain and low-level background wind climatology compared to those in Gobi Desert (Chen et al., 2017; Liu et al., 2019), Taklimakan Desert is more representative to study the effects of dust aerosol solar radiative forcing on local region than the Gobi Desert because of its huge dust emission capability (Chen et al., 2017).

Kashi represents a place heavily affected by dust aerosol particles. It is influenced by local anthropogenic pollution and pollution transported from surrounding arid and desert areas. According to the SONET long-term measurements from 2013,

the Kashi site is frequently affected by dust, where the multi-year average *AOD* is up to 0.56±0.18 at 500 nm. Moreover, the Ångström exponent (*AE*, 440~870 nm) and fine-mode fraction (*FMF*, 500 nm) at Kashi are the lowest (with the multi-year average values of 0.54±0.27 and 0.40±0.14, respectively, low values of *AE* indicate the presence of large dust particles) among

all 16 sites within SONET around China (Li et al., 2018). In contrast, the multiyear average *AOD*s (500 nm) at Zhangye (0.28±0.11) and Minqin (0.26±0.11) are only half of that at Kashi or less, meanwhile, their average values of *AE* and *FMF* are also greater than those at Kashi (Li et al., 2018). These data imply that coarse particles are more dominant in the Taklimakan Desert in comparison with the Gobi Desert. Every year, *FMF* reaches the lowest value, and the volume particle size distribution presents a pre-dominant coarse mode from March to May at Kashi (Li et al., 2018), due to the frequent dust invasions in spring.

Chen et al. (2014) also reported that the dust radiative forcing had relatively small inter-annual variation, but a distinct seasonal course with maximum values in late spring and early summer during the period of 2007 to 2011 in the Taklimakan Desert. Sun et al. (2012) found that the solar radiative heating peaks appear in April in southern Xinjiang and in May for northern Xinjiang. Thus, the DAO-K intensive field campaign was carried out in April 2019 and lasted for nearly a month. During the campaign, several dust events were observed on the base of a coordinated deployment of multiple in-situ and remote sensing

platforms and state-of-the-art instruments based on passive and active detection technologies.

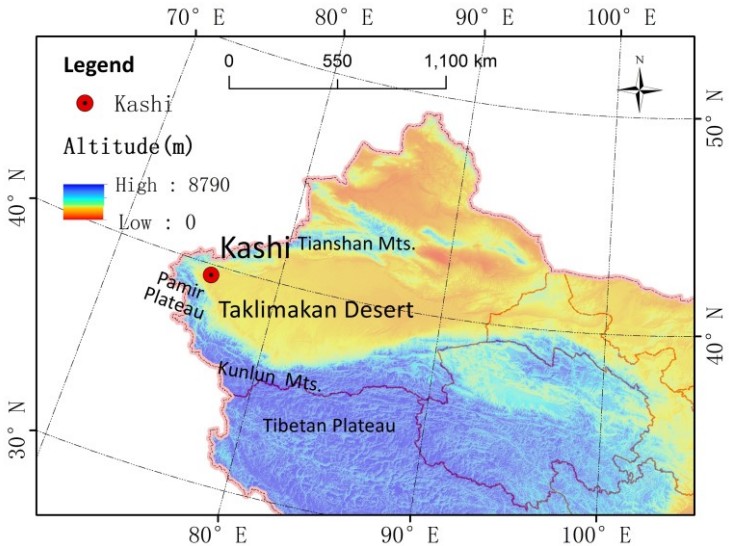

**Figure 1: The location of the observation site (Kashi) of the DAO-K field campaign.**

**2.2 Instrumentation**

Columnar aerosol properties are essential parameters for quantifying radiative forcing of atmospheric aerosol particles. However, high loading and complex light scattering processes corresponding to diverse particle shapes bring challenges to remote sensing of mineral dust in the atmosphere (Dubovik et al., 2006; Bi et al., 2010; Li et al., 2019). Ground-based detection by sun-sky radiometer works out a solution by modelling dust particles as randomly oriented spheroids in the retrieval

framework (Dubovik et al., 2006). From these activities, quality-assured databases of dust aerosol properties became available based on both of AERONET and SONET sun-sky ratiometer retrievals (Holben et al. 1998; Li et al., 2018). During the DAO-K campaign, four Cimel sun-sky radiometers, including a polarized sun-sky-moon radiometer CE318-TP (#1150), two unpolarized sun-sky-moon radiometers CE318-T (#1098 and #1141), and a polarized sun-sky radiometer CE318-DP (#0971), were deployed at Kashi (Fig. 2a). CE318 #1150 and #1141were calibrated rigorously at the AERONET Izaña Observatory with the accuracy of $AOD$ about 0.25 %~0.5 %, while $AOD$-related measurements and sky radiance measurements of CE318 #1098 and #0971 were calibrated via the master instrument #1150 by a vicarious/transfer calibration method before the field campaign (Holben et al. 1998; Li et al., 2008; 2018). The volume aerosol parameters of $AOD$, $SSA$, $AE$, and asymmetry factor (i.e., $g$) in four channels with center wavelengths of 440, 675, 870, 1020 nm were retrieved following the SONET level 1.5 data criteria (Li et al., 2018). Observations from the CE318 #1141 also joined in the AERONET dataset. The consistency of the products following the AERONET and SONET retrieval frameworks has been validated by Li et al. (2018). The multi-wavelength properties of $AOD$, $SSA$, $AE$, and $g$ were applied in RT simulations. In addition to sun-sky radiometers, a METONE BAM-1020 Continuous Particulate Monitor was also deployed to measure $PM_{2.5}$ mass concentration (mg m$^{-3}$) (Fig. 2a). The hourly $PM_{10}$ mass concentration (mg m$^{-3}$) were collected from the routine measurements of ambient air quality continuous automated monitoring system in Kashi operated by China National Environmental Monitoring Center. The aerosol parameters including $AOD$, $PM_{2.5}$ and $PM_{10}$ mass concentrations were assimilated in WRF-Chem model simulation in this study.

Aerosol radiative effects also depend on the surface albedo and the vertical structure of atmosphere (Wendisch et al., 2004). During the DAO-K campaign, atmospheric profiles, including the vertical distributions of the atmospheric pressure, temperature, and relative humidity, were collected from sounding balloon measurements. The sounding balloons were operationally launched twice a day around 0:00 UTC and 12:00 UTC at Kashi weather station (39.46˚N, 75.98˚E, 1291 m above mean sea level). Normally, about 3000 individual measurements are recorded during one balloon flight, which corresponds to a sampling frequency of 1 second (Guo et al., 2016; Chen et al., 2019). The data quality was controlled following the operational specifications for conventional upper-air meteorological observations (China Meteorological Administration, 2010). The accuracy of the temperature profile in the troposphere is within ±0.1 K (Zhang et al., 2018; Guo et al, 2019). In addition to pressure, temperature, and relative humidity profiles, ozone profiles obtained by the Ozone Monitoring Instrument (OMI)/Aura satellite (Bhartia et al., 1996) were used as input for the RT model. The satellite observations of the Moderate resolution imaging spectroradiometer (MODIS)/Terra+Aqua were employed to collect the surface reflection during the DAO-K campaign. The MODIS products of shortwave bidirectional reflectance distribution function (BRDF) parameters, black-sky albedo (BSA), and white-sky albedo (WSA) were adopted to derive the surface albedo during daytime (Schaaf and Wang, 2015). A solar radiation monitoring station, equipped with an EKO MS-57 pyrheliometer and two MS-80 pyranometers, was used for measuring the direct, diffuse, and total solar irradiances (W m$^{-2}$) in the range of 0.28~3.0 μm (Fig. 2a). The pyrheliometer and pyranometers have been calibrated before the campaign with uncertainties of 0.55 % and 0.66 %, respectively. They satisfy the requirements of class A under the ISO 9060:2018 with response time of less than 0.2 and 0.4

seconds, separately. The fraction of diffuse skylight radiation deduced from the diffuse and total irradiances also gave a key weighting index to modulate the diurnal-changes of the surface albedo.

Further instruments provided independent evidence of the existence of dust and cloud layers during the observations.
Multiwavelength Mie-Raman polarization lidar (LILAS) developed by the Laboratoire d'Optique Atmosphérique, Université de Lille 1 (Fig. 2b), was equipped with three elastic wavelengths (all linearly polarized) at 355, 532, 1064 nm and three Raman wavelengths at 387, 530, 408 nm, from which the vertical distribution of multiple optical and physical properties of dust aerosol particles can be obtained (Veselovskii et al., 2016, 2018; Hu et al., 2019). The backscattering coefficient profile at 355 nm wavelength was applied in this study to distinguish the two-layer structure of dust. The YNT all sky view camera ASC200
equipped with two wide-dynamic full-sky visible and infrared imagers, recorded dynamic states of clouds during day and night with 10 min (or less than 10 min) resolution. An overview of the instruments and corresponding parameters employed in the study is listed in Table 1. Considering different durations of various measurements, we calculated and discussed the *ASRF* from 2 to 25 April 2019, when simultaneous measurements are available.

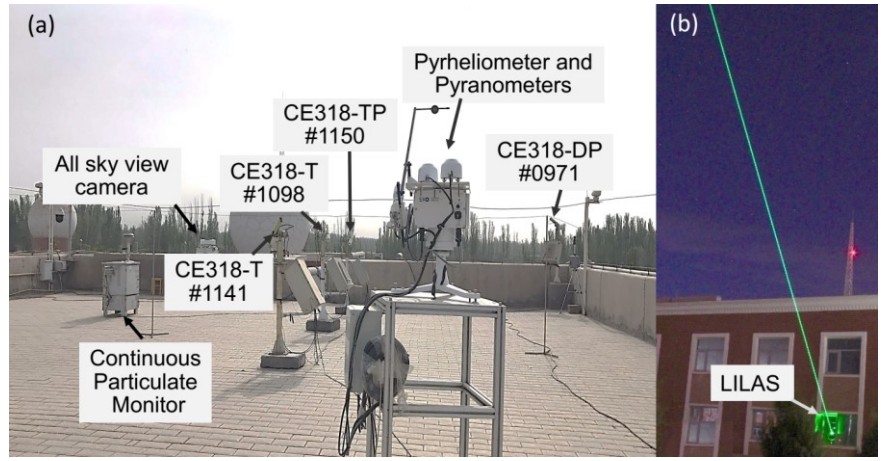

**Figure 2: Setup of experimental apparatus of the DAO-K field campaign (a) on the roof, (b) in door.**

**Table 1: Overview of the parameters and instruments employed in the radiative transfer and WRF-Chem model simulations and validation.**

| Application | Parameter | Instrument | Time period of operation |
|---|---|---|---|
| Radiative transfer simulation | **aerosol properties** aerosol optical depth Ångström exponent single scattering albedo asymmetry factor | sun-sky radiometer | 1/4/2019~25/4/2019 |
| | **atmospheric profiles** vertical distributions of atmospheric pressure, | sounding balloon | 1/4/2019~30/4/2019 |

| | | | |
|---|---|---|---|
| | temperature,<br>relative humidity | | |
| | Ozone profile | OMI/Aura | 1/4/2019~30/4/2019 |
| | **land surface albedo** | | |
| | shortwave BRDF parameters<br>shortwave black-sky albedo<br>shortwave white-sky albedo | MODIS/Terra+Aqua | 1/4/2019~30/4/2019 |
| | diffuse solar irradiance<br>total solar irradiance | pyranometers | 2/4/2019~28/4/2019 |
| WRF-Chem simulation | aerosol optical depth | sun-sky radiometer | 1/4/2019~25/4/2019 |
| | $PM_{2.5}$ mass concentration | continuous particulate monitor | 1/4/2019~28/4/2019 |
| | $PM_{10}$ mass concentration | ambient air quality continuous automated monitoring system | 1/4/2019~30/4/2019 |
| Evidences and validation | direct normal solar irradiance<br>diffuse solar irradiance<br>total solar irradiance | pyrheliometer<br>pyranometers | 2/4/2019~28/4/2019 |
| | backscattering coefficient | LILAS | 4/4/2019~28/4/2019 |
| | full-sky visible image | all sky view camera | 2/4/2019~27/4/2019 |

## 3 Estimation of aerosol solar radiative forcing

### 3.1 Definition of aerosol solar radiative forcing

The direct solar radiative forcing of atmospheric aerosol particles is calculated using the following equations (Babu et al.,2002; Adesina et al., 2014; Esteve et al., 2014):

$$ASRF_{\text{TOA}} = F_{\text{net,TOA}}^{\text{a}} - F_{\text{net,TOA}}^{0}, \tag{1}$$

$$ASRF_{\text{BOA}} = F_{\text{net,BOA}}^{\text{a}} - F_{\text{net,BOA}}^{0}, \tag{2}$$

$$ASRF_{\text{ATM}} = ASRF_{\text{TOA}} - ASRF_{\text{BOA}}, \tag{3}$$

$$F_{\text{net}} = F^{\downarrow} - F^{\uparrow}, \tag{4}$$

where $ASRF_{TOA}$, $ASRF_{BOA}$ and $ASRF_{ATM}$ denote the direct aerosol solar radiative forcing at the TOA, BOA, and in ATM, respectively. $F_{net}^{a}$ and $F_{net}^{0}$ indicate the net irradiances with and without aerosols, respectively. $F^{\downarrow}$ and $F^{\uparrow}$ separately represent the downward and upward irradiances. All the above quantities are measured in physical units of W m$^{-2}$. The radiative forcing efficiency is defined as the rate at which the atmosphere is forced per unit of aerosol optical depth at 550 nm (García et al., 2008, 2012):

$$ASRFE = ASRF \Big/ \tau_{550} , \tag{5}$$

where $ASRFE$ (in W m$^{-2}$ $\tau_{550}^{-1}$) is the aerosol solar radiative forcing efficiency at the TOA, BOA, or in ATM. Since the effects of aerosol loading on radiative forcing have been eliminated, radiative forcing efficiency has unique advantage on evaluation of the direct radiative effects of different types of aerosols (García et al., 2008).

## 3.2 Radiative transfer simulations

The focus of this study is to quantify the direct $ASRF$ and $ASRFE$ at the TOA, BOA, and in ATM under cloud-free conditions using the SBDART model fed with comprehensive ground-based and satellite observations collected during the DAO-K campaign. SBDART is a radiative transfer software tool that has been widely applied in atmospheric radiative energy balance studies (Ricchiazzi et al.,1998; Li et al., 2018). The discrete ordinate method is adopted in the code, which provides a numerically stable algorithm to solve the equations of plane-parallel radiative transfer in a vertically inhomogeneous atmosphere (Ricchiazzi et al.,1998). The simulations cover the same wavelength range (i.e., 0.28~3.0 µm) as the pyranometer for convenience of comparison. Simulations of the $ASRF$ by the SBDART model are susceptible to the input conditions including the aerosol properties, atmosphere profiles, and land surface albedo. These input data were specified based on the high-quality dataset obtained in the DAO-K campaign.

### 3.2.1 Aerosol properties

The aerosol properties including $AOD$, $SSA$, $AE$, and $g$ were retrieved from the radiometer observations at four bands with the central wavelengths at 440, 675, 870, and 1020 nm. They were applied in the instantaneous radiative forcing and efficiency calculations at the corresponding observing time. The aerosol properties in the SW range are obtained by interpolation and extrapolation using parameters in the above mentioned four wavelength bands. For daily mean $ASRF$ simulation, the averaged aerosol parameters (i.e., $AOD$, $SSA$, $AE$, and $g$) obtained from the day-time radiometer observations were used as alternatives of the daily mean aerosol properties. The daily mean aerosol radiative forcing and efficiency were calculated by taking the average of the 24 instantaneous values on an hourly basis.

### 3.2.2 Atmospheric profiles

In addition to aerosol properties, atmospheric profiles of thermodynamic properties are important for the *ASRF* calculations. The vertical distributions of air pressure, temperature, water vapor, and ozone densities exert obvious influences on the direct and diffuse solar irradiances at the BOA. The predefined atmospheric profiles in the used RT model (e.g., tropical, mid-latitude summer, mid-latitude winter, sub-arctic summer, sub-arctic winter profiles) are different from Kashi local conditions. Therefore, within the *ASRF* simulations, the predefined profiles have been replaced by the actual measurements conducted

during the DAO-K campaign. Vertical distributions of the atmospheric pressure, temperature, relative humidity can be obtained by atmospheric sounding twice a day around 0:00 UTC and 12:00 UTC at Kashi. The profiles of ozone density (in g m$^{-3}$) were deduced from the OMI/Aura OMO3PR product (in DU) (Bhartia et al., 1996). Two atmospheric profiles were specified for each day. The profile closest to the *ASRF* simulated moment was adopted for both of instantaneous and daily mean aerosol radiative forcing estimates.

### 3.2.3 Surface albedo

Land surface albedo (LSA) is another key factor to influence the radiation budget, mainly due to its significant impact on the SW upward irradiance (Liang, 2004; Wendisch et al., 2004; Bierwirth et al., 2009; Tegen et al., 2009; Jäkel et al., 2013; Stapf et al., 2020). Shortwave land surface albedo $\alpha_{SW}$, also known as blue-sky albedo, can be calculated from the black-sky albedo $\alpha_{SW}^{BSA}$ and white-sky albedo $\alpha_{SW}^{WSA}$ weighted by the fraction of diffuse skylight radiation (Schaaf et al., 2002; Wang et al.,

2015):

$$\alpha_{SW}(\theta_s, \varphi_s) = f_{diffuse,SW}\ \alpha_{SW}^{WSA} + (1 - f_{diffuse,SW})\ \alpha_{SW}^{BSA}(\theta_s, \varphi_s), \tag{6}$$

where $f_{diffuse,SW}$ denotes the fraction of diffuse radiation in the solar spectral range. $(\theta_s, \varphi_s)$ specifies the incident solar geometry (i.e., solar zenith angle and solar azimuthal angle).

The shortwave WSA and BSA are provided by the MODIS BRDF/Albedo Science Data Product MCD43A3, which is

225 produced daily using 16 days of MODIS/Terra+Aqua data. MCD43A3 only delivers the surface albedo products at local solar noon. However, diurnal variations of LSA cannot be ignored, which has been demonstrated by previous studies (Lewis and Barnsley, 1994; Lucht et al., 2000; Wang et al., 2015). There will be an obvious bias in estimating daily solar radiation when simply using the local noon value as a surrogate of daily mean albedo (Wang et al., 2015). As for the weighting parameters of the RossThickLiSparseReciprocal BRDF model (i.e., isotropic, volumetric, and geometric), the changes within 16 days are

230 subtle. Therefore, the daily three model weighting parameters over the SW band afforded by the MODIS product MCD43A1 are adopted to derived the WSA and BSA (the latter is as a function of incident solar direction) at different *ASRF* simulated moments. The fraction of diffuse radiation can be calculated by the ratio of diffuse solar irradiance to total solar irradiance,

which mainly depends on the solar zenith angle, aerosol, and cloud conditions. The diffuse and total irradiances measured by pyranometers with 1 min resolution are applied in this study to calculate the fraction of diffuse radiation.

Fig. 3 illustrates the diurnal variations of LSA and corresponding full-sky visible images under four typical sky conditions at Kashi. For the cloud-free and low aerosol loading conditions (identified as clear sky, e.g., almost the whole day of 7/4/2019 and afternoon of 12/4/2019), LSA changes distinctively for different time. High values of LSA are observed in the early morning and the late afternoon. Meanwhile, the extreme value of LSA in the morning (0.253) is greater than that in the afternoon (0.218), which has been supported by some other field observations (Minnis et al., 1997; Wang et al., 2015). The

local noon albedo shows very low value. The daily mean albedo under the clear-sky condition (0.199) is significantly greater than the local noon albedo (0.173). However, in dust-polluted (almost the whole days of 9/4/2019 and 25/4/2019) and cloudy (the morning of 12/4/2019) sky conditions, the changes of LSA are not as severe as in the clear-sky conditions. Nevertheless, the local noon albedo still cannot reflect the effects of aerosol and cloud variations on land surface albedo. Thus, diurnal-changed LSA and the daily mean albedo were adopted in the instantaneous and daily mean *ASRF* simulations, respectively. It

is expected that estimations of instantaneous and daily mean aerosol radiative forcing can be improved by considering diurnal variations of LSA instead of local noon albedo.

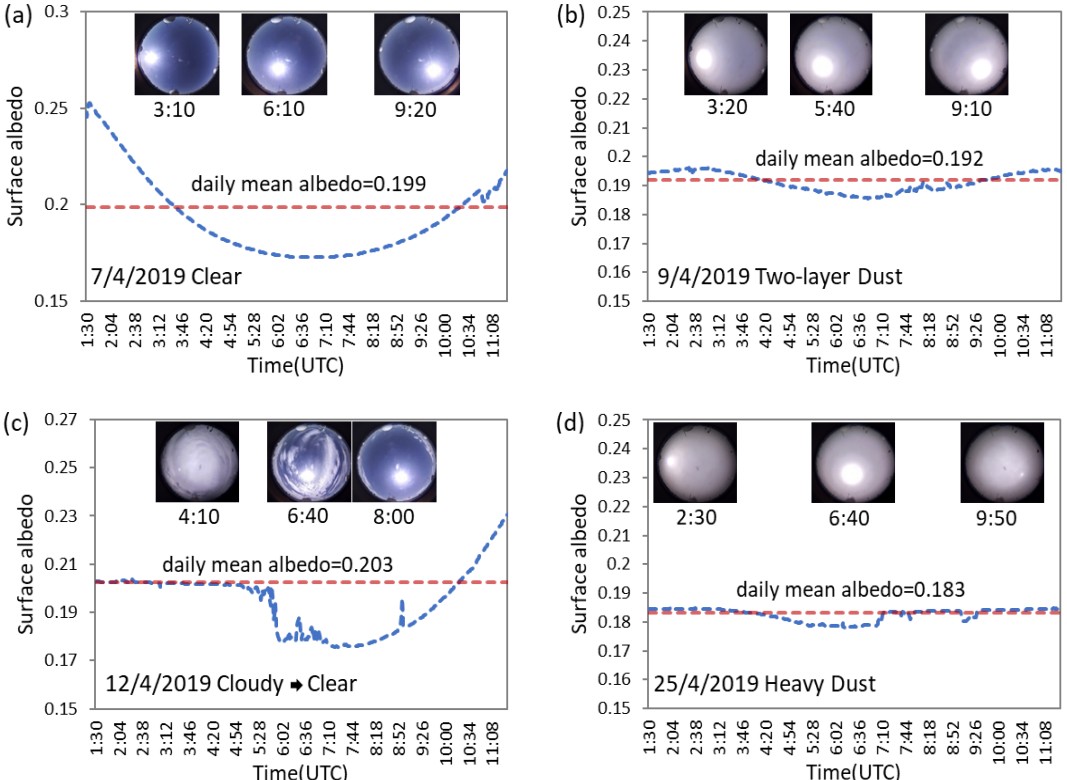

Figure 3: Diurnal variations of blue-sky albedo and corresponding full-sky visible images under different sky conditions at Kashi (a) clear case, (b) two-layer dust case, (c) clouds early/clearing late case, (d) heavy dust case.

### 3.3 WRF-Chem simulations

 ### 3.3.1 Forecast model

The Weather Research and Forecasting model with chemistry (WRF-Chem) model version 4.0 (Grell et al., 2005; Fast et al., 2006) was used to simulate the *ASRF* at Kashi. The simulations were configured in a 9 km domain centered at the Kashi site with 45×45 grid points and 41 vertical levels that extended from the surface to 50 hPa. The main physical options used for this study included the Purdue Lin microphysics scheme, the unified Noah land surface model, the Yonsei University (YSU) scheme for planetary boundary layer meteorological conditions, and the Rapid Radiative Transfer Model for General Circulation Models (RRTMG) for solar and terrestrial radiation (Lin et al., 1983; Mlawer et al., 1997; Chen and Dudhia, 2001; Hong et al., 2006; Iacono et al., 2008). The Carbon Bond Mechanism (CBMZ) was used for the Gas-phase chemistry processes (Zaveri and Peters, 1999), which included aqueous-phase chemistry. The aerosol chemistry was based on the Model for Simulating Aerosol Interactions and Chemistry (MOSAIC; Zaveri et al., 2008) with four size bins (0.039~0.156 μm, 0.156~0.625 μm, 0.625~2.5 μm, 5.0~10.0 μm dry diameters). The sum of aerosol mass concentrations in the first three size bins constructs the concentration of $PM_{2.5}$ and the sum of the four size bins gives the concentration of $PM_{10}$. Aerosol types such as sulfate, methanesulfonate, nitrate, ammonium, black carbon, primary organic carbon, sodium, calcium, chloride, carbonate, aerosol liquid water, and other inorganic matter (e.g., trace metals and silica) are involved in the simulation. Dust was simulated with the GOCART dust emission scheme (Ginoux et al., 2001). The dust particulates were aggregated into the other inorganic matter component and were presented in the calculation of aerosol optical properties with anthropogenic aerosols.

Aerosol particle optical properties were calculated as a function of wavelength based on the Mie theory. The aerosol components within each size bin are assumed to be internally mixed. The mixing refractive indices are the volume-weight average in refractive indices of all aerosol components. Aerosol extinction and scattering coefficients and asymmetry factors for a particulate per size bin are attained though searching a look-up Mie table by Chebyshev polynomial interpolation with the desired mixing refractive indices and wet particulate radius. The value of particulate extinction coefficient multiplied with the particulate number concentration is volume extinction coefficient which is then multiplied with the height of layer to attain the layer *AOD* value. The sum of all layer *AOD* values over the four size bins is the columnar total *AOD* and is used for calculating *AOD* increments in the assimilation.

 ### 3.3.2 Assimilation system

The Gridpoint Statistical Interpolation (GSI) 3DVAR assimilation system version 3.7 was applied to improve the simulated aerosols by assimilating the aerosol measurements collected at Kashi during the DAO-K campaign (Wu et al., 2002; Kleist et al., 2009). This GSI version has been modified to assimilate the aerosol products (Liu et al., 2011b; Schwartz et al., 2012). We assimilated our ground-based multi-wavelength *AOD* (440, 675, 870, 1020 nm) and the surface-layer concentrations of $PM_{2.5}$ and $PM_{10}$ suited to the MOSAIC aerosol module in WRF-Chem. We used the natural logarithm of particulate number

concentration per size bin as control variables. The aerosol dry mass concentrations, particulate number concentrations and aerosol water contents are converted into *AOD* per size bin using the WRF-Chem aerosol optical routine. The adjoint observation operators for *AOD* and particulate matter are given as

$$\frac{\delta \ln(\tau)}{\delta \ln(n_i)} = \frac{n_i}{\tau} \cdot \frac{\delta \tau}{\delta n_i} = \frac{n_i}{\tau} \cdot e_i = \frac{\tau_i}{\tau},$$  (7)

$$\frac{\delta \ln(c)}{\delta \ln(n_i)} = \frac{n_i}{c} \cdot \frac{\delta c}{\delta n_i} = \frac{n_i}{c} \cdot r_i,$$  (8)

where $n_i$ is aerosol number concentration in the *i*th size bin, $\tau$ and $c$ are the observed *AOD* and particulate matter mass concentrations. As no aerosol particle extinction coefficient assimilated in this experiment, we assume the extinction coefficient per size bin is constant in grid at each model layer. Innovation of number concentration due to *AOD* constraint is therefore a proportion of change in model layer *AOD* to the observed columnar *AOD*, which is attained via iteration to minimize the cost function. Innovation of number concentration due to the constraints of *PM*$_{2.5}$ and *PM*$_{10}$ are associated with the ratios ($r_i$) of mass concentrations to number concentrations in a size bin estimated in the guess field, weighted by the proportion of the size number concentration, changing in the iteration, to the total particulate matter concentration.

### 3.3.3 Model setup

Initial and lateral boundary conditions for the meteorological fields in the WRF-Chem simulations were generated from the National Centers for Environmental Prediction (NCEP) Final Analysis (FNL) data using the Global Forecast System (GFS) model at a horizontal resolution of 1°. The boundary conditions were updated every 6 hours and then interpolated linearly in time by WRF-Chem. Anthropogenic emissions from the 2010 MIX emission inventories (www.meicmodel.org) containing the Multi-resolution Emission Inventory of China (MEIC) were used in the simulations. The biogenic emissions were estimated using the Model of Emissions of Gases and Aerosols from Nature (MEGAN; Guenther et al., 2006). Two one-month WRF-Chem simulations were performed for April 2019, discarding a one-week spin-up at the beginning of each simulation. The first one-month simulation was used for modelling background error covariance. The second one-month simulation was assimilated the observations of *PM*$_{2.5}$, *PM*$_{10}$ and *AOD* with GSI at 0:00, 6:00, 12:00 and 18:00 UTC with the assimilation window of ±3 h centered at the analysis time. The model was restarted from the meteorology and chemistry at analysis time and ran to the next analysis time. For the second one, each restart called the radiation routines twice which included and excluded the aerosols, respectively, and the corresponding difference between the two calls in irradiances is aerosol radiative forcing.

A general way to model background error covariance is the National Meteorological Center (NMC) method that computes the statistical differences between two forecasts with different leading lengths (e.g. 12 and 24 h, or 24 and 48 h) but valid at the same time (Parrish and Derber, 1992). However, in some experiments, the WRF-Chem model underestimated aerosol concentrations and hence likely lowered the error magnitudes. For this reason, we assessed the standard deviations of the

control variables over the entire one-month period at the four analysis hours (i.e., 0:00, 6:00, 12:00 and 18:00 UTC), respectively. Each standard deviation field was used for modelling a background error covariance repeatedly applied in the assimilations at the corresponding analysis hour. This approach represents the strong fluctuations of control variables as weather evolution during clear and dusty days. We expect fluctuations of aerosols over the different weathers are larger than the uncertainties due to different leading forecast lengths and may give a better input field for modelling background error covariance. The observation errors for *AOD* and *PM* were 50 % of natural logarithm of 0.01 and those errors of *PM* including measurement error and representative error depending on the grid size and the *PM* concentrations (Schwartz et al., 2012). The choice of 50 % was determined by trying experimentally with different values, which can effectively assimilate measurements and will not excessively damage the model results.

## 4 Results of radiative transfer simulations

### 4.1 Aerosol solar radiative forcing and efficiency

The time series of the measured values of *AOD*, *AE*, $PM_{2.5}$ and $PM_{10}$ mass concentrations collected during the DAO-K campaign are shown in Fig. 4. The average value of *AOD* at 550 nm wavelength is 0.65 during the campaign. According to *AOD*, five high aerosol loading episodes are identified: UTC 9:26~12:15 on 2/4/2019, 9:13 on 3/4/2019 until 5:11 on 5/4/2019, 1:52 on 8/4/2019 until 4:20 on 10/4/2019, 1:47 on 13/4/2019 until 12:32 on 16/4/2019, 1:30 on 24/4/2019 until 4:11 on 25/4/2019. The highest values of *AOD* at 550 nm (2.3) were observed from 24/4/2019 to 25/4/2019 during a severe dust storm event. From Fig. 4, a negative correlation between *AOD* and *AE* becomes obvious. For the five high aerosol loading episodes, the *AE*s show very low values, suggesting that the heavy aerosol outbreaks at Kashi were dominated by dust particles. As a qualitative indicator of aerosol particle size, the values of *AE* are always less than 1.0 during the DAO-K campaign, illustrating the fact that aerosol particles around the Taklimakan Desert are mainly dominated by coarse particles (even for clear situations). This is consistent with the results obtained in a previous study (Fig. 4 in Li et al., 2018). Comparatively high values of *AE* (>0.4) are observed on 7, 12, 19, and 23 April 2019, implying relatively small particle enrichments for these days. The time series of $PM_{2.5}$ and $PM_{10}$ mass concentrations generally concur with that of *AOD*. However, for some days, such as 19 and 23 April 2019, relatively high $PM_{2.5}$ corresponding to low *AOD* has been observed, indicating an enhanced influence of anthropogenic pollutions. For the measurements on 7 and 12 April 2019, high *AE* values corresponding to low $PM_{2.5}$ concentrations could be down to the very low turbidity conditions. It should be noted that the errors in computations of *AE* significantly increase under low aerosol loading conditions (Kaskaoutis et al., 2007).

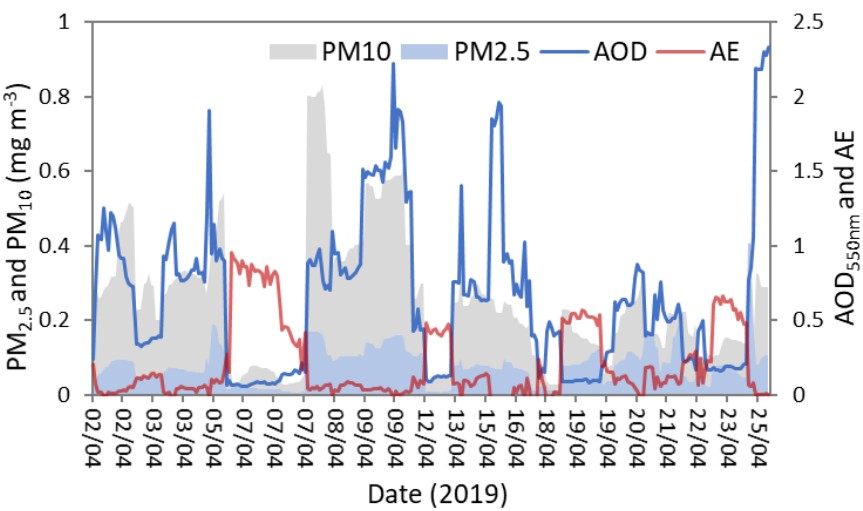

**Figure 4: Variations of aerosol optical depth (550 nm), Ångström exponent (440~870 nm), *PM*₂.₅ and *PM*₁₀ mass concentrations at Kashi site during the DAO-K campaign.**

Results of instantaneous *ASRF* and *ASRFE* during the DAO-K campaign are given in Fig. 5. Both positive and negative values of *ASRF*, corresponding to warming and cooling effects respectively, can be found at top of the atmosphere (Fig. 5a). However, aerosols have only warming effects in the atmosphere (Fig. 5c) and cooling effects at the surface (Fig. 5e) during the DAO-K campaign. *ASRF* values at the TOA and BOA exhibit obvious negative correlations with *AOD*. Positive correlation are observed between *ASRF* within the atmospheric column and *AOD*. From Fig. 5, it is evident that the dust aerosol has strong influences on the solar radiation budget. For the five high aerosol loading episodes (Fig. 4), the dust-dominant aerosol population exerts stronger cooling effects at the TOA and BOA, and more significant warming effects in the atmosphere than other low aerosol loading situations. Moreover, the cooling effects at the BOA are more noticeable than at the TOA, with minimum values around -217 W m⁻² and -119 W m⁻², respectively.

When *ASRF* is normalized by aerosol optical depth at 550 nm wavelength, the *ASRFE* is obtained. This quantity is mostly insensitive to the aerosol loading, at least if a linear relation between *ASRF* and *AOD* is assumed. Nevertheless, a weak negative correlation between *ASRFE* and *AE* can be observed at the BOA (Fig. 5f). That means, the *ASRFE* at the surface can roughly indicate the radiative forcing effects of different types of aerosols (García et al., 2008). Relatively large fraction of small particles associated with high *AE* has stronger *ASRFE* for cooling the surface than other low *AE* situations. But for TOA and ATM (Fig. 5b, d), there is no obvious correlation between *ASRFE* and *AE*. Generally, the cooling effect of aerosols at Kashi is more efficient at the BOA than that at the TOA. It is in accordance with the results of *ASRF*. In comparison with *ASRF*, the variation of *ASRFE* is relatively moderate during the campaign. The strongest cooling effects on the TOA and BOA all appear in the episode of dust storm outbreak (i.e., 24 and 25 April 2019) (see Fig. 5a, e). But large dust particles in this case do not show extreme radiative forcing efficiency (Fig. 5 b, f). Strong cooling efficiencies at the surface during the DAO-K campaign occur in the very clear cases with high *AE* on 7 April 2019 (Fig. 5f).

During the DAO-K campaign, the average values of daily mean $ASRF$ at Kashi are $-19\pm13$ W m$^{-2}$ at the TOA and $-36\pm23$ W m$^{-2}$ at the BOA, which are slightly stronger than the multiyear average values at this site (i.e., -16 W m$^{-2}$ at the TOA and -18 W m$^{-2}$ at the BOA) obtained by the previous study (Li et al., 2018). These results are reasonable, since the campaign was performed in the dust-prone season and higher aerosol loading situations have stronger $ASRF$ effects as discussed above. Likewise, the average values of daily mean $ASRFE$ at the TOA and BOA during the DAO-K campaign are $-27\pm9$ W m$^{-2}$ $\tau_{550}^{-1}$ and $-55\pm10$ W m$^{-2}$ $\tau_{550}^{-1}$, respectively, which are more efficient than the corresponding multiyear average values (i.e., -21 W m$^{-2}$ $\tau_{550}^{-1}$ at the TOA and -24 W m$^{-2}$ $\tau_{550}^{-1}$ at the BOA) reported in the previous study (Li et al., 2018).

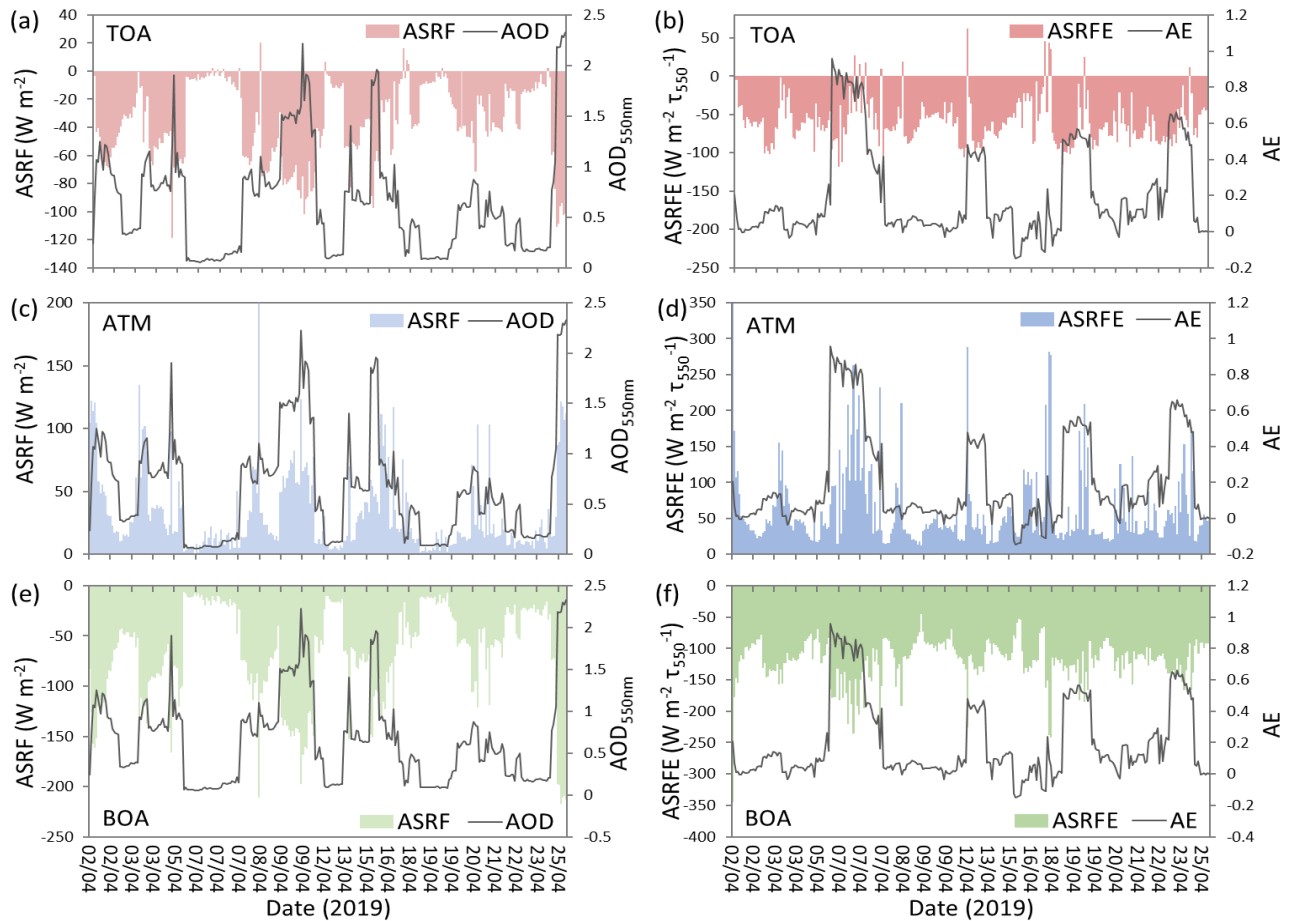

**Figure 5: Instantaneous aerosol solar radiative forcing (left column) and efficiencies (right column) at Kashi site during the DAO-K campaign (upper panels: TOA; middle panels: ATM; lower panels: BOA).**

### 4.1.1 Clear-sky case

Instantaneous $ASRF$ and $ASRFE$ of the clear-sky case on 7 April 2019 is depicted in Fig. 6. It was a typical cloud-free and low aerosol loading day at Kashi with $AOD$ at 550 nm less than 0.22 for the whole day. As discussed above, the highest $AE$ is observed on this day during the one-month campaign (see Fig. 4). Both cooling and warming effects of aerosols can be found

at the top of atmosphere. The cooling effects of *ASRF* are up to -19 W m$^{-2}$ at the TOA and -48 W m$^{-2}$ at the BOA, and the warming effect of *ASRF* is up to 50 W m$^{-2}$ in the atmosphere. The corresponding extreme *ASRFE* values are -126, -236, and 263 W m$^{-2}$ $\tau_{550}^{-1}$, respectively It is apparent that the changes of *ASRFE* are more intense than the corresponding *ASRF* for the clear case.

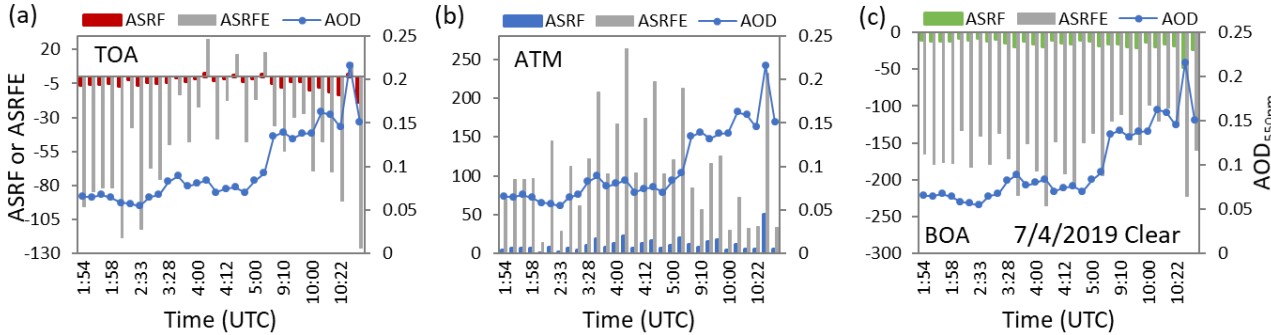

**Figure 6: Instantaneous aerosol solar radiative forcing and efficiencies of the clear-sky case on 7 April 2019 at Kashi site (a) TOA, (b) ATM, (c) BOA.**

### 4.1.2 Heavy dust case

Fig. 7 describes *ASRF* and *ASRFE* for a heavy dust storm episode on 25 April 2019 at Kashi. Only few observations from 3:33 to 4:11 UTC were suitable for retrieval in this day. Aerosol optical depth at 550 nm was up to 2.3 during this observation

period. In comparison with the clear case, dust particles have stronger cooling effects at the TOA and BOA (*ASRF*s up to -111 and -217 W m$^{-2}$, respectively), and stronger warming effect in ATM (*ASRF* up to 121 W m$^{-2}$). However, we observe the extreme *ASRFE* values of -51, -99, and 55 W m$^{-2}$ $\tau_{550}^{-1}$ at the TOA, BOA, and in ATM, respectively, indicating that the radiative forcing of dust is less efficient than that of the clear case. Moreover, the variations of *ASRFE* in the dust case are more moderate than which of *ASRF*. These are strikingly differences from the clear-sky case.

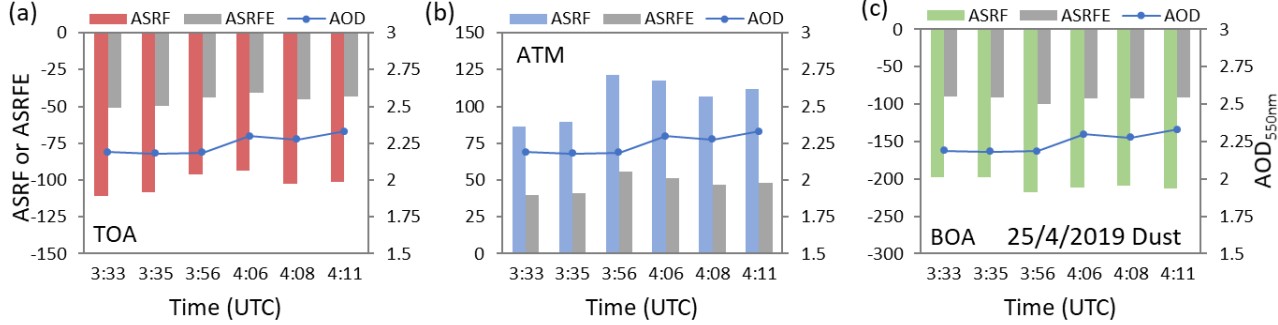

**Figure 7: As Fig. 6, but for the heavy dust case on 25 April 2019.**

### 4.1.3 Two-layer dust case

On 9 April 2019 one extra layer suspending above the planetary boundary layer (PBL) was observed. Fig. 8 illustrates the observations of LILAS on 8 April. Lidar observations on 9 April 2019 are not shown because the lidar stopped working due to technical problems in the night of 8 April 2019. According to the backscattering coefficient profiles at 355 nm, the lower layer and upper layer can be clearly identified. Lidar measurements indicate that aerosols in the layer above the PBL are probably dust particles because the derived high depolarization ratios agree with the values for dust. However, from lidar measurements we cannot draw unambiguous conclusion about the aerosol type in the PBL, because the incomplete overlap range of the lidar system is up to 800~1000 m. From Fig. 4, high *AOD* corresponding to low *AE* in the whole atmosphere and high $PM_{2.5}$ and $PM_{10}$ concentrations in the surface layer are exhibited from 8 to 9 April. It also suggests the complex pollutions by two-layer dust particles during this pollution process. *AOD* at 550 nm on 9 April changes from 1.4 to 2.2 (Fig. 9). In consistent with the above heavy dust case, only cooling effects can be observed at the TOA and BOA, and only warming effect can be found in ATM for this case. The two layers of dust particles result in a TOA cooling effect up to -102 W m$^{-2}$, BOA cooling effect of up to -198 W m$^{-2}$, and atmosphere warming effect of up to 123 W m$^{-2}$. The absolute values of *ASRF* at the TOA and BOA in this case are all less than those in the heavy dust case, suggesting the aerosols in the heavy dust case have more powerful cooling effects. Nevertheless, the extreme values of *ASRFE* are -62, -105, and 58 W m$^{-2}$ $\tau_{550}^{-1}$ at the TOA, BOA and in ATM, respectively, indicating that dust particles in the two-layer case have stronger radiative forcing efficiencies than those in the heavy dust cases.

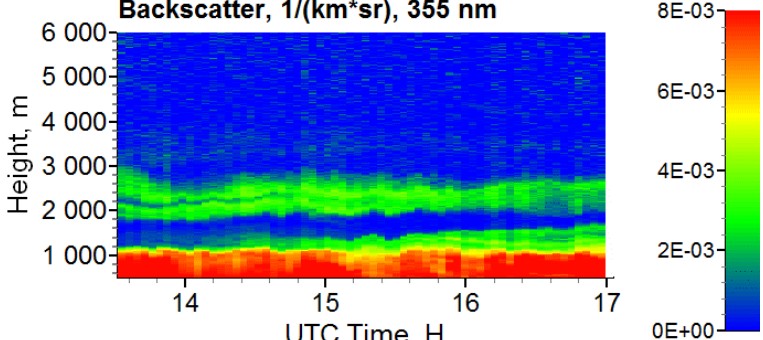

**Figure 8: The backscattering coefficient profiles at 355 nm for the two-layer dust case in the night of 8 April 2019.**

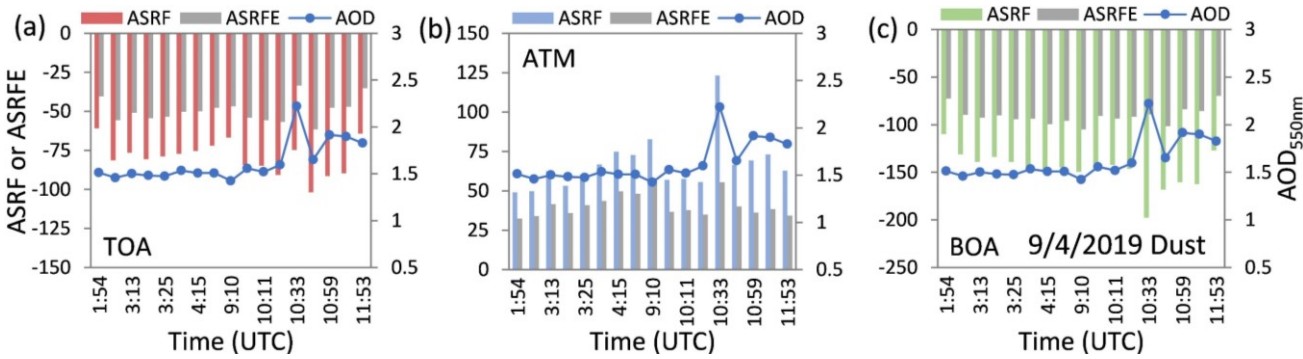

Figure 9: As Fig. 6, but for the two-layer dust case on 9 April 2019.

## 4.2 Influences of the atmosphere and surface conditions

Fig. 10 describes the influences of atmospheric profile and land surface albedo on the simulations of total irradiances and *ASRF*. The differences in the results of total downward irradiance (*TDI*), total upward irradiance (*TUI*), and *ASRF* at the TOA and BOA simulated with the pre-defined midlatitude winter profile and user-specified profiles, and simulated with local noon surface albedo and instantaneous surface albedo are given, respectively. According to Fig. 10a, different settings of profiles have no influence on the *TDI* at the TOA. For the *TUI*, the absolute differences are less than 9 W m$^{-2}$. However, the atmospheric profile has significant impacts on both the *TDI* and *TUI* at the surface. The influences on *TDI* are generally stronger than which on *TUI*. The maximum absolute difference is up to 138 W m$^{-2}$ (Fig. 10c). For *ASRF* at the TOA, the effects of atmospheric profiles are less than 5 W m$^{-2}$. But the serious influences can up to 103 W m$^{-2}$ on *ASRF* at the BOA (Fig. 10e). The average effect of different profiles on *ASRF* is 0.8 W m$^{-2}$ at the TOA, which is quite small in comparison with the average values of daily *ASRF* (-19 W m$^{-2}$). However, the average difference of 13 W m$^{-2}$ for *ASRF* affected by atmospheric profiles cannot be ignored relative to the average *ASRF* (-36 W m$^{-2}$) at the BOA. As a result, the cooling effects of aerosol radiative forcing will be significantly underestimated at the BOA simulated with the pre-defined midlatitude winter profile instead of the user-specified Kashi atmospheric profiles.

Like atmospheric profile, different settings of LSA have also no influence on *TDI* at the TOA (Fig. 10b). They have small effects on *TDI* at the BOA (absolute difference less than 3 W m$^{-2}$), but obvious impacts on *TUI* at the TOA and BOA (absolute difference up to 22 W m$^{-2}$) (Fig. 10 b, d). From Fig. 3, the local noon albedo is often less than the daily mean albedo. Especially for the clear day, the minimum of LSA occurs around the local noon. Then the *TUI* at the TOA and BOA will generally be underestimated by using the local noon albedo instead of instantaneous surface albedo in the simulations. But for *ASRF* (Fig. 10f), two LSA settings lead to moderate impacts at the TOA and BOA with average absolute differences of 1.8 and 1.7 W m$^{-2}$, respectively. Therefore, simulations using the local noon albedo trend to overestimate the cooling effects of the aerosol radiative forcing both at the TOA and BOA.

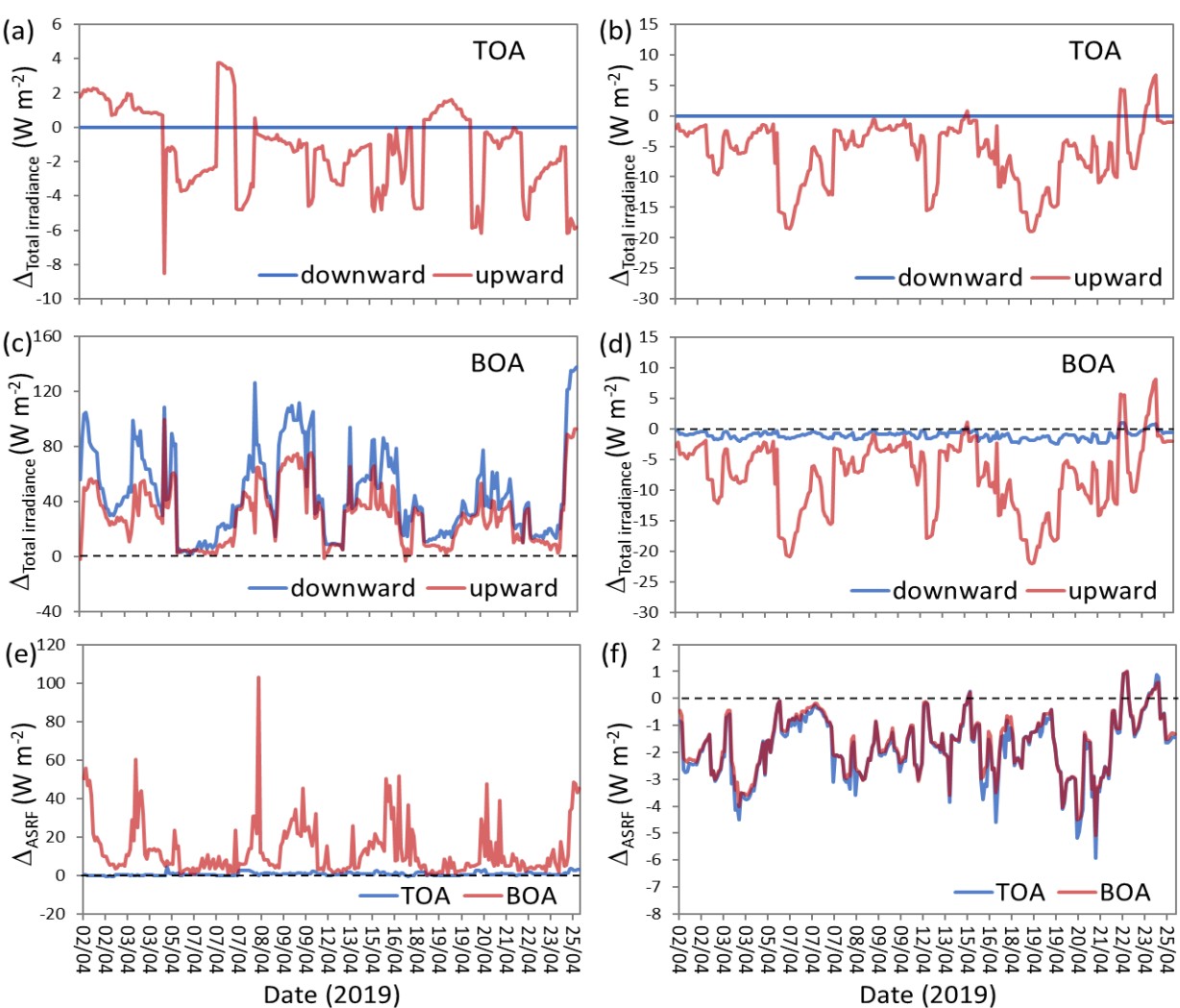

Figure 10: Influences of atmospheric profile (left column) and land surface albedo (right column) on total irradiances and *ASRF*. (a) differences of total downward and upward irradiances (*TDI* and *TUI*) at the TOA between simulations with the pre-defined midlatitude winter profile and user-specified profiles; (b) differences of *TDI* and *TUI* at the TOA between simulations with local noon surface albedo and instantaneous surface albedo; (c) as (a), but for BOA; (d) as (b), but for BOA; (e) differences of *ASRF* between simulations with the pre-defined midlatitude winter profile and user-specified profiles at the TOA and BOA; (f) differences of *ASRF* between simulations with local noon surface albedo and instantaneous surface albedo at the TOA and BOA.

## 4.3 Difference from AERONET products

Aerosol radiative forcing at the TOA and BOA are operational products provided routinely by AERONET. Measurements of the CE318 #1141 during the DAO-K campaign have been processed by AERONET. Therefore, we can compare the *ASRF* product from AERONET with our simulations. For AERONET, broadband upward and downward irradiances in the SW ranges from 0.2 to 4.0 μm were calculated by radiative transfer model with retrieved aerosol properties as model inputs (http://aeronet.gsfc.nasa.gov). However, AERONET adopts different definition of *ASRF* that only taking the downward

irradiance at the BOA and the upward irradiance at the TOA into consideration (García et al., 2012). The upward irradiances with and without aerosols in Eq. (2), along with the downward irradiances with and without aerosols in Eq. (1), are not taken into account. Omitting the downward irradiances will not make much difference in *ASRF* at the TOA. But for *ASRF* at the

BOA, it is predictable that neglecting the upward irradiance will lead to obvious difference. Some existing studies have executed this kind of comparison (García et al., 2008; García et al., 2012; Bi et al., 2014) and reported that AERONET trends to overestimate aerosol *ASRF* at the BOA (García et al., 2012).

Fig. 11 presents the correlations of instantaneous aerosol *ASRF* between the RT model simulations and the AERONET products. It is obvious that there are linear relationships between our RT simulations and the AERONET results with $R^2$ up to

0.98 and 0.99 at the TOA and BOA, respectively. Two *ASRF* results at the TOA show good consistency with a slope of 1.01, even though the calculated SW ranges are not exact match (i.e., 0.28~3.0 μm for this study, and 0.2~4.0 μm for AERONET). But for BOA, the AERONET products are obvious stronger than the corresponding RT model simulations (with a slope of 1.24), which agrees with the conclusion of the previous study (García et al., 2012).

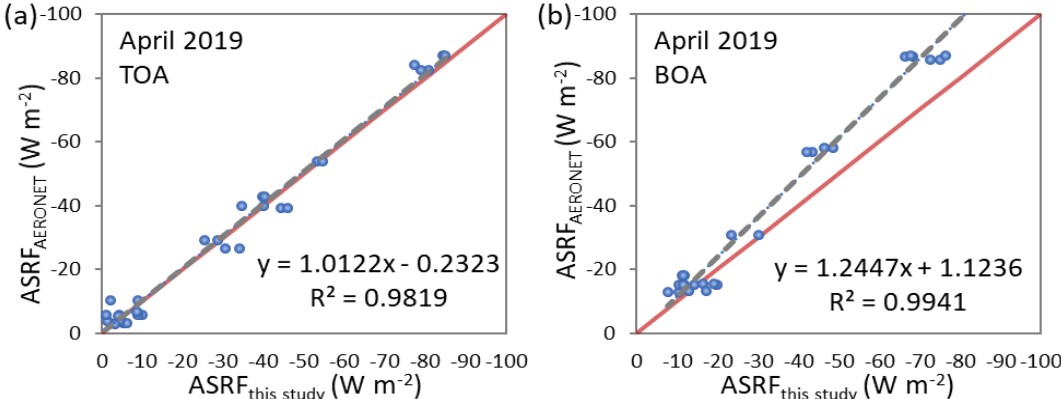

**Figure 11: Correlations of instantaneous *ASRF* between radiative transfer (RT) model simulations in this study and the AERONET products during the DAO-K campaign (a) TOA, (b) BOA.**

## 5 Comparison with WRF-Chem simulations

### 5.1 Comparison between radiative transfer and WRF-Chem simulations

Fig. 12 compares the assimilated aerosols to the observations. Evidently, the assimilation greatly improves the particulate

matter concentrations and show reasonable variations in accordance with the dust episodes. However, two disadvantages are noticeable. One is the assimilation fails to reproduce the extremely high $PM_{2.5}$ and $PM_{10}$ on 24~25 April 2019, because the background error covariance is not specific for the model error in the strong dust storm. A better model result for the specific dust storm requires improving the model capability of simulating dust emission and the transport of dust particulates besides data assimilation. Another is the assimilated *AOD* indeed increases but not well approaches the observations. The reason is

that we only assimilated *AOD* by assuming the invariable extinction coefficients. Hence, this low bias in *AOD* cannot be

eliminated by choosing a scaling factor smaller than 50 % in the observation error for that it will damage the surface-layer particulate results.

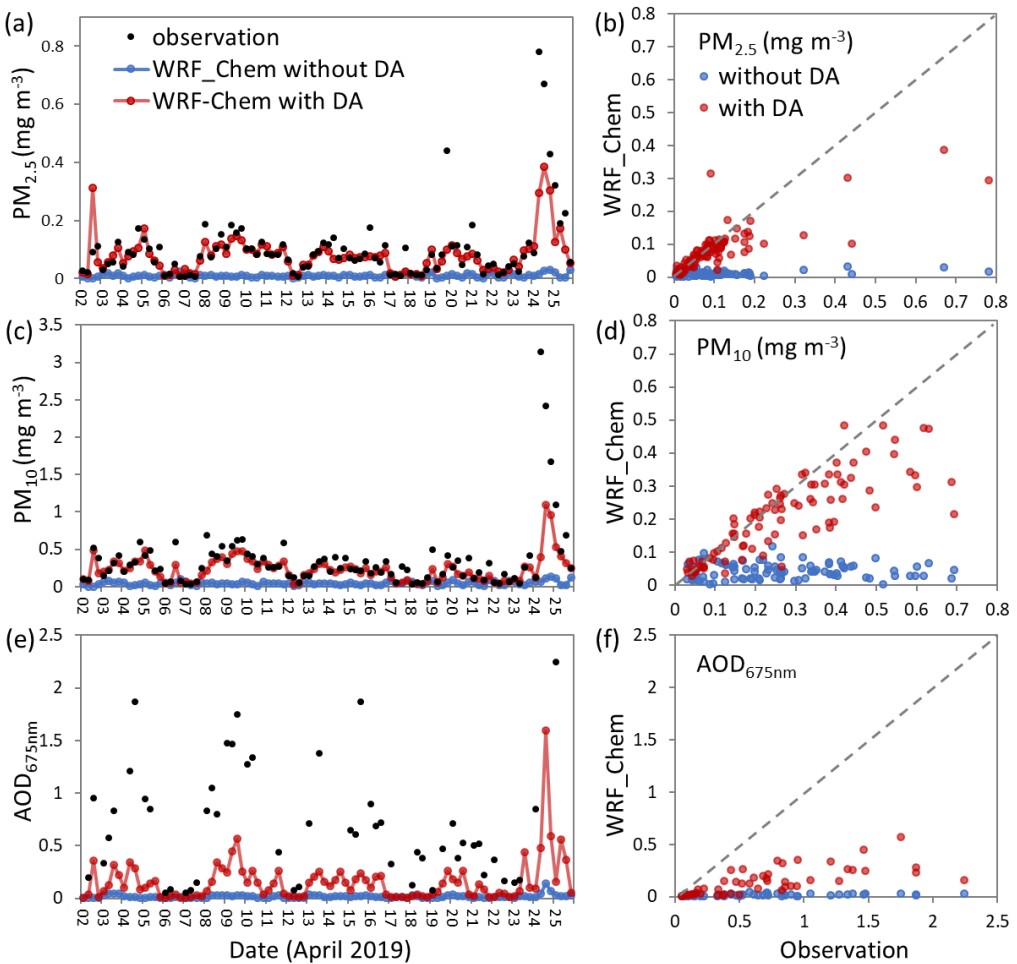

**Figure 12: Comparisons of the surface-layer *PM*$_{2.5}$ (a, b), *PM*$_{10}$ (c, d) concentrations and *AOD* at 675 nm (e, f) among the observations, the WRF-Chem simulations with and without data assimilations (DA) in April 2019. The observations have been interpolated to 0:00, 6:00, 12:00, 18:00 UTC of each day.**

Fig. 13 illustrates the results of daily mean *ASRF*s during DAO-K campaign simulated by the SBDART and WRF-Chem models. Two results show similar variation patterns. However, there are obvious differences between the WRF-Chem results and the RT simulations in some dust-polluted cases (e.g., 9, 24, and 25 April 2019). According to the RT simulations, the strongest radiative forcing occurred on 25 April 2019. However, the most significant *ASRF* of WRF-Chem simulation is found on 24 April 2019 followed by 25 April 2019. As mentioned above, heavy dust storms broke out on these two days during the DAO-K campaign. The percent differences are sometimes greater than 50 % between the RT and WRF-Chem simulations. The significant differences between the two kinds of simulated results should be further evaluated.

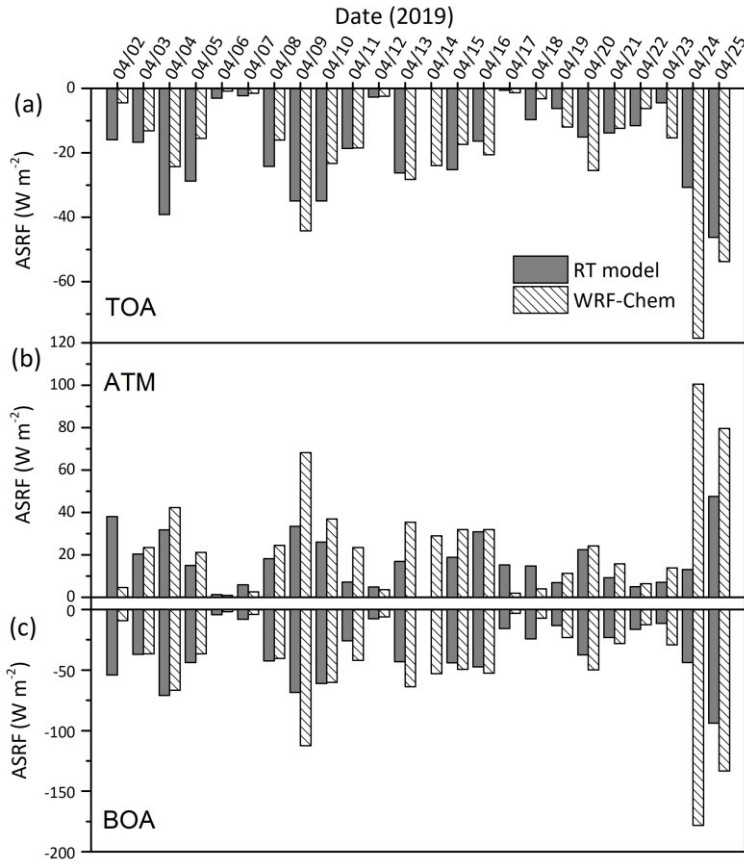

**Figure 13: Comparisons of daily mean *ASRF* between the RT model calculations and the WRF-Chem simulations during the DAO-K campaign (a) TOA, (b) ATM, (c) BOA.**

## 5.2 Validation by ground-based irradiance measurements

Fig. 14 directly compares the RT and WRF-Chem simulated downward irradiances at surface with the ground-based measurements under three different sky conditions (i.e., clear case, heavy dust case, and two-layer dust case). The RT simulations of total, direct, and diffuse downward irradiances in the three situations agree well with high-precision measurements of pyrheliometer and pyranometers. The percent differences of RT-simulated total irradiance with respect to the measurements are only 0.03 % for the clear case, -2.67 % for the heavy dust case, and -0.43 % for the two-layer dust case. Except for the heavy dust case, they are within the pyranometer measurement uncertainties (0.66 %). As for the WRF-Chem simulations, the total irradiances in the clear-sky case are consistent with RT simulations and measurements (Fig. 14a). But for the direct irradiances, there are obvious differences between the WRF-Chem simulations and the corresponding measurements (Fig. 14b). Moreover, the WRF-Chem simulated diffuse irradiances in the clear case (Fig. 14c), the total, direct, and diffuse irradiances in the heavy dust and two-layer dust cases (Fig. 14d~i) are significantly distinct from the measurements and RT simulations.

One of the most noticeable features in the curves of WRF-Chem results is the sudden jump around 6:00 UTC, which can
be attributed to data assimilations restarted at 6:00 UTC and ran to the next analysis time 12:00 UTC. The WRF-Chem results
are greatly improved after 6:00 UTC in the dust-polluted cases. It is evident that data assimilations can ameliorate the WRF-
Chem simulations in dust cases, but the correction effects are still limited. So, the problems of the WRF-Chem simulation have
not yet been fully resolved by the assimilations of aerosol optical depth and particulate matter concentrations. This conclusion
is in accordance with Figs. 12 and 13. Our measurements have proved that the simulations of RT model are reliable in both of
clear and high aerosol loading situations. The WRF-Chem model preforms better in clear sky than in the dust-polluted
conditions. There is still room for improving the WRF-Chem simulation of dust aerosol radiative forcing.

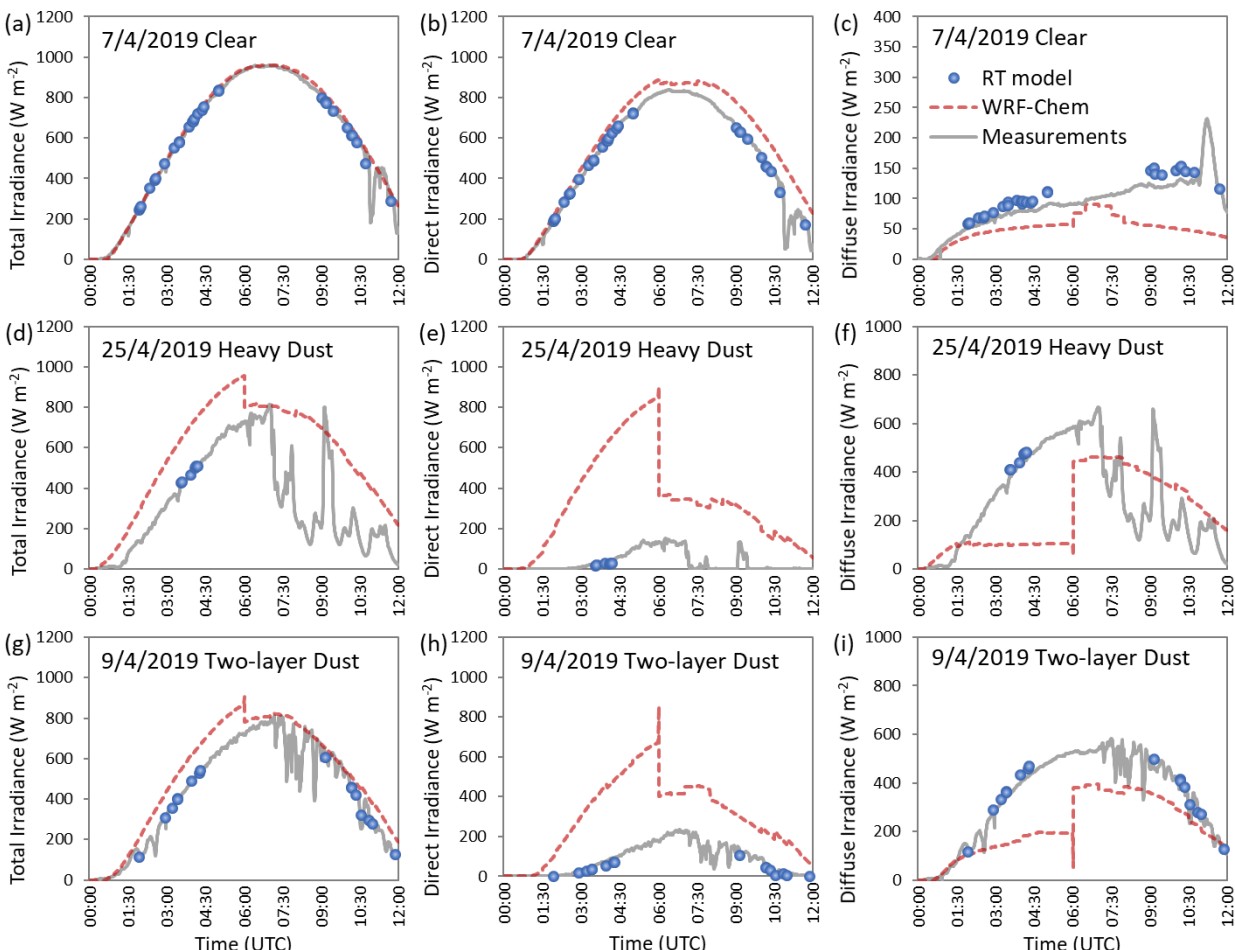

**Figure 14: Comparisons of total, direct and diffuse downward irradiances at the bottom of atmosphere for the clear-sky case (upper three panels), the heavy dust case (middle three panels), and the two-layer dust case (lower three panels) at Kashi site (blue points:**
**simulated by the RT model; red dash lines: simulated by the WRF-Chem model with data assimilations at 0:00 and 6:00 UTC; gray solid lines: measured by pyrheliometer and pyranometers).**

# 6 Summary and conclusions

Dust aerosol particles play an important role in local and global climate changes by influencing the solar radiation budget through scattering and absorbing processes, especially for the region close to dust sources such as deserts. The complicated scattering and absorption characteristics of dust particles make it challenging to estimate their direct radiative forcing. To overcome some of the issues with the quantification of the dust radiative effects, the Dust Aerosol Observation-Kashi (DAO-K) campaign was designed and preformed near the Taklimakan Desert, which represents a substantial and stable source of Asian dust aerosol particles. For almost one month, comprehensive observations of aerosol properties (i.e., aerosol optical depth, Ångström exponent, single scattering albedo, and asymmetry factor), atmospheric profiles (including ozone measurements), and land surface properties were obtained by a variety of state-of-the-art ground-based and satellite instruments in the dust season, and were applied to estimate the aerosol solar radiative forcing using the SBDART radiative transfer model. In addition to high-quality datasets of volumetric aerosol properties, satisfying the AERONET and SONET level 1.5 data criteria, the daily specified atmospheric profiles and diurnal variations of surface albedo were also considered in the calculations. The results simulated with the SBDART model show that the average values of daily mean *ASRF* at Kashi are -19 W m$^{-2}$ at the TOA and -36 W m$^{-2}$ at the BOA during the DAO-K campaign. The dust-dominant aerosol particles have stronger cooling effects at both the TOA and BOA, and more significant warming effects in the atmosphere than other low aerosol loading situations. Nevertheless, the radiative forcing efficiencies in dust-polluted cases exhibit lower than those in clear-sky conditions. The average influences of different profiles on *ASRF* are small at the TOA (0.8 W m$^{-2}$) but remarkable at the BOA (13 W m$^{-2}$). The cooling effects of aerosol radiative forcing at the BOA is significantly underestimated by simulations with the pre-defined midlatitude winter profile instead of the user-specified profiles measured at Kashi during the DAO-K campaign. Simulations using the local noon albedo trend to overestimate the cooling effects of the aerosol radiative forcing both at the TOA and BOA. Different land surface albedo settings (i.e., local noon albedo or instantaneous albedo) lead to moderate impacts on *ASRF* with average effects of 1.8 W m$^{-2}$ at the TOA and 1.7 W m$^{-2}$ at the BOA.

By assimilating the multi-wavelength columnar *AOD* and the surface-based measurements of $PM_{2.5}$ and $PM_{10}$ mass concentrations, the aerosol solar radiative forcing was also simulated for the time period of DAO-K field campaign using the WRF-Chem model. The measurements of downward solar irradiances at the surface were used as reference in evaluating the RT and WRF-Chem simulations. The direct, diffuse (and the sum of both) downward irradiances simulated by the SBDART model in the clear-sky, heavy dust, two-layer dust conditions are all in sufficient agreement with ground-based measurements. As for the WRF-Chem simulations, the total irradiances in the clear-sky case are consistent with RT calculations and measurements. But the direct, diffuse, and total irradiances simulated by WRF-Chem significantly deviate from measurements in the dust-polluted situations. Based on these findings it is concluded that the SBDART model provides credible estimates of dust particle solar radiative forcing if supplied with appropriate model input data. Data assimilations can obviously improve the WRF-Chem simulations in dust cases, but the correction effects are still limited. Considering the actual measured atmospheric profiles and diurnal cycles of land surface albedo has some potential to improve the RT simulations. Optimizations

of dust emission scheme, background error setting of dust assimilation system, dust parameterization including nonsphericity, are proposed as promising approaches to improve the WRF-Chem simulations of dust radiative forcing. We would like to emphasize, however, that in this study the comparisons are conducted at one site and in a limited time period. Future research on this topic should include a systematic evaluation of RT and WRF-Chem simulations on extended space and time scales.

*Data availability.* The MODIS, OMI, and AERONET products can be accessed at https://modis.gsfc.nasa.gov/, https://disc.gsfc.nasa.gov/, and https://aeronet.gsfc.nasa.gov/, respectively (last accessed July 2019).

*Author contributions.* ZL, PG, LL, KL, and JW designed the Dust Aerosol Observation-Kashi (DAO-K) campaign. YO and CL conducted the measurements of the solar radiation monitoring station and the all sky view camera. YO collected and processed the data of atmospheric profiles. QH performed the Lidar observation. The retrievals of aerosol properties were processed and provided by KL. The WRF-Chem simulations and analysis were provided by WC. LL improved the SBDART simulations and conducted data analysis and comparisons. LL and MW prepared the paper with contributions from all authors.

*Competing interests.* The authors declare that they have no conflict of interest.

*Acknowledgements.* This research was funded by the National Natural Science Foundation of China (NSFC), grant number 41871271, and the National Key R&D Program of China, grant number 2016YFE0201400. The authors acknowledge the groups of MODIS, OMI, and AERONET for making the surface albedo, ozone profile, and radiative forcing products available, respectively. We also thank the Kashi regional meteorological bureau and the China National environmental monitoring centre for providing the data of atmospheric sounding and $PM_{10}$ concentration of Kashi, respectively. The Anhui Yunnengtian Intelligent Technology Co., Ltd., China is acknowledged for providing the All sky view camera and technical support. The first author also wishes to thank Haofei Wang, Thierry Podvin, Igor Veselovskiy, Jie Chen, and Ying Zhang for participating the measurements. We are grateful to the anonymous reviewers whose valuable comments and suggestions that have helped us to improve the paper.

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
