# Peer review of "Aerosol solar radiative forcing near the Taklimakan Desert based on radiative transfer and regional meteorological simulations during the Dust Aerosol Observation-Kashi campaign"

_Atmospheric Chemistry and Physics, 2020_

## Referee Comment (RC1) · Anonymous Referee #1 · 11 Mar 2020

General comments: In this manuscript, the measurements obtained during the Dust Aerosol Observation-Kashi campaign were employed in radiative transfer model and the estimations were improved by considering the actual measured atmospheric profiles and diurnal variations of land surface albedo. Direct aerosol solar radiative forcing of dust aerosols was analyzed based on comprehensive parameters and numerical models. The effects of data assimilations on estimating the radiative forcing effects were also explored. However, the manuscript was poorly worded, thus making me confused. The manuscript needs to structure writing accurately to produce proper paragraphs with clear topics. Major revisions are necessary before the manuscript is finally accepted for publication. Specific comments: 1.Section 1, this part should intro-

duce the research background and significance, current status, concealed problems, as well as research mentality and content of this manuscript. Nevertheless, the introduction of this manuscript is inundated with accumulation of literatures rather than sublimation of these research results. The authors need survey more literatures in recent five years and then summarized them. 2.Section 2 and 3, the authors seem to be drowned by abundant resources and avoid stringing them together to form a system. Some descriptions should be streamlined. The outline and structure of this manuscript should be reorganized. 3.Section 2.1, this part should explain why the experimental site was selected in Kashi instead of the local aerosol properties, such as the representativeness or speciality in studying aerosol-related issues. 4.Lines 85-86, 'the Ångström exponent (AE, 440∼870 nm) and fine-mode fraction (FMF) at Kashi are the lowest among all sites in China' What is the scientific value of this sentence? And it needs strong literature to support. 5.Section 2.2, 'aerosol properties during the DAO-K campaign' is part of the 'Results', so I suggest moving it to Section 4. 6.The structure of the manuscript makes me feel that some parts are more or less irrelevant to the title 'Solar radiative forcing of aerosol particles near the Taklimakan desert during the Dust Aerosol Observation-Kashi campaign in Spring 2019'. Too much attention was spent on Section 4.3. Technical corrections: 1.Line 17, 'are improved by' should be changed to 'were improved by'. 2.Line 40, 'it is a challenging' should be changed to 'it is challenging' or 'it is a challenge'. 3.Line 41, add 'the' before 'high surface albedo over desert'. 4.Line 42, I suggested replacing the sentence 'Numerous efforts have investigated. . .' with 'Numerous efforts have been undertaken to investigate. . .'. 5.Line 52, 'have relatively small inter-annual variation' should be changed to 'had relatively small inter-annual variation'. 6.Line 53, 'According to WRF-Chem simulations' should be changed to 'According to the WRF-Chem simulations'. 7.Line 85, the comma before 'moreover' should be changed to semicolon. 8.Line 239, 'includes' should be changed to 'included'. 9.Line 315, 'Globally' should be changed to 'Generally'. 10.Line 433, 'for it will damage the surface-layer particulate results' should be changed to 'for that it will damage the surface-layer particulate results'. 11.I suggest deleting some acronyms,

especially the phrases only appear once. Too many acronyms make the article chaotic.

---

## Referee Comment (RC2) · Anonymous Referee #2 · 14 Mar 2020

The dust aerosol has great impact on weather and climate both regional and global scales. However, quantifying this impact remains elusive, mainly owing to the lack of full consideration of dust aerosol properties, atmospheric profiling, and surface albedo. Li et al attempted to address the scientific question by calculating the ASRF by means of comprehensive observations, which was further compared with WRF-Chem and AERONET products. Overall, the manuscript is well structured, the methods are technically sound, and the main findings presented seem to be reasonable and be of general interests to the aerosol and climate fields. I think the topic fits within the scope of ACP. I would recommend acceptance of this MS for publication pending the following revisions:

**Major comments:**

1. P2L35-38: This sentence is confusing to me. If my understanding is right, it is meant to express the dust originated from Taklimakan Desert (TD) exerts influences the air quality and climate over the downstream regions via long-range transport. Therefore, please try to be specific instead of using general words. However, some key references are missing, since both observations (Liu et al., 2019, doi:10.1029/2019GL083508.) and model simulations (Chen et al., 2017, doi: 10.1007/s11430-016-9051-0) suggested that the dusts generated in TD have LESS impacts on downstream regions due to the unique terrain and low-level background wind climatology, compared with those from other deserts in northwestern China.

2. Figures 3, 5, 10, 12: The X-axis can be considered to be revised (more minor ticks and labels are needed to be given), given the ASRF and ASRFE are only able to be estimated during daytime without clouds. Another important issue is the cloud-induced impact on the radiation reaching the surface. The authors are better to analyze the day-by-day variation of cloud (fraction) over the study sites of Kashi, which is concurrent with the ground-based aerosol remote sensing and radiation observations. I believe this will provide more insights into the community of aerosol radiative forcing.

3. Figure 11: The readers would like to know how the ASRF is derived from AERONET, instead of the performance of ASRF product. The details will shed light on the difference between ASRF from RT model and ASRF from AERONET.

**Minor comments:**

Abstract: What are the two simulations in "The percent difference of daily mean ASRF between the two simulations." ? which is supposed to be described specifically.

P2L34: The dust aerosol originated from western China was revealed to exert significant impact on the mesoscale convection in downwind regions such as North China (Li et al., 2017, doi: 10.1038/s41598-017-12681-0), which exemplified well the dynamic effect of dust. Therefore, this reference can be considered to be added here.

P2L34-35: Recent studies also show that the dust RF strongly depends on the overlapping pattern of dust aerosol and cloud layer in the vertical. Therefore, this sentence might as well be revised to "The dust radiative effects also depend on the surface albedo over the desert (Bierwirth et al., 2009) and the underlying clouds as well (Waquet et al., 2013, doi: 10.1002/2013GL057482; Xu et al., 2017, doi: 10.1016/j.atmosenv.2017.07.036)"

P2L50: "were used" should be revised.

P2L58-61: what does "the modulate effects" mean? Besides, it seems strange in "performances of models…validated by comparing with the observations of AOD.." . I guess it is supposed to mean that ASRF from model …validated against that incorporating the observations of AOD…. Please clarify it or make modifications to them.

P3L84: Please specify the years in "more than six years".

P3L85-85: it needs some references to support this statement "…the lowest among all sites in China. " . it really depends on the stations you refer to. e.g., the aerosol properties at Tazhong should be dominated by dust aerosol if you have observations therein.

P5L122-123: More details are needed for the sounding measurements, including the launching time and location, sampling resolution, data uncertainties, e.t.c. Reference support is required.

P7L149-151: I am confused again for the descriptions shown here are not consistent with those in Fig. 3. For instance, "The maximum $PM_{10}$ concentration ..from 24 to 25 April 2019 was up to 4 mg m$^{-3}$" cannot be derived from Fig. 3. Also, "no CE318 measurement around the peak time of dust outbreak." disagreed with continuous AOD curves.

---

## Author Comment (AC2) · 23 May 2020

**To Anonymous Reviewer #2: (Comment ID:acp-2020-60-RC2)**

Dear Reviewer,

Thank you very much for your detailed and supportive comments. You raised important issues in the comments. They can help us to improve the manuscript to a better scientific level. We have carefully taken these comments into consideration and have revised our manuscript accordingly. Please find below the comments in blue italics and our responses in black and the changes in bold.

Responses to major comments:

*1. P2L35-38: This sentence is confusing to me. If my understanding is right, it is meant to express the dust originated from Taklimakan Desert (TD) exerts influences the air quality and climate over the downstream regions via long-range transport. Therefore, please try to be specific instead of using general words. However, some key references are missing, since both observations (Liu et al., 2019, doi:10.1029/2019GL083508.) and model simulations (Chen et al., 2017, doi: 10.1007/s11430-016-9051-0) suggested that the dusts generated in TD have LESS impacts on downstream regions due to the unique terrain and low-level background wind climatology, compared with those from other deserts in northwestern China.*

Reply: We thank the reviewer for pointing out these issues around influences of dust aerosol particles originated from Taklimakan Desert. In response to the comments, we have rephrased this sentence in a more specific way. The suggested references have been added. Please find them in the second paragraph of section "1 Introduction" in the revised manuscript:

"**As one of the largest sandy deserts in the world, the Taklimakan Desert located in the Xinjiang Uygur Autonomous Region of China is a main source region of Asian dust (Huang et al., 2009), which influences not only surrounding areas such as the Tibetan Plateau (Liu et al., 2008; Chen et al., 2013; Yuan et al., 2019), but also wide regions in Eastern Asia (Mikami et al., 2006; Liu et al., 2011b; Yuan et al., 2019), even North America and Greenland through long-range transports across the Pacific Ocean (Bory et al., 2003; Chen et al., 2017; Liu et al., 2019).**"

The issue of less impacts of Taklimakan dust particles on downstream regions due to the unique terrain and low-level background wind climatology, and the reason that the experimental site was

selected near Taklimakan desert instead of the Gobi desert were also referred and discussed. Please find them in the subsection "2.1 Observation site" in the revised manuscript:

[revised manuscript text omitted]

*2. Figures 3, 5, 10, 12: The X-axis can be considered to be revised (more minor ticks and labels are needed to be given), given the ASRF and ASRFE are only able to be estimated during daytime without clouds. Another important issue is the cloud-induced impact on the radiation reaching the surface. The authors are better to analyze the day-by-day variation of cloud (fraction) over the study sites of Kashi, which is concurrent with the ground-based aerosol remote sensing and radiation observations. I believe this will provide more insights into the community of aerosol radiative forcing.*

Reply: **More minor ticks and labels of the x-axes have been added in Figs. 4 (old Fig. 3), 5, 10**. For Fig. 12, in order to compare the WRF-Chem simulations and observations point by point, we interpolated the observations of $PM_{2.5}$, $PM_{10}$, and $AOD$ at 675 nm to the corresponding assimilation time of 0:00, 6:00, 12:00 and 18:00 UTC of each day. Please also see the statement "The second one-month simulation was assimilated the observations of $PM_{2.5}$, $PM_{10}$ and $AOD$ with GSI at 0:00, 6:00, 12:00 and 18:00 UTC with the assimilation window of ±3 h centered at the analysis time." in the first paragraph of subsection "3.3.3 Model setup". So, the ticks and labels were given with daily intervals in Fig. 12. For each day, there are four sets of $PM_{2.5}$, $PM_{10}$ in Figs. 12a and c. But normally less than four sets of $AOD$ can be obtained owing to cloud screening or other quality control operation (see Fig. 12e). **To clarify this, the caption of Fig. 12 has been changed into: "Comparisons of the surface-layer $PM_{2.5}$ (a, b), $PM_{10}$ (c, d) concentrations and $AOD$ at 675 nm (e, f) among the observations, the WRF-Chem simulations with and without data assimilations (DA) in April 2019. The observations have been interpolated to 0:00, 6:00, 12:00, 18:00 UTC of each day.**"

We thank the comment on cloud-induced impacts which raises an important issue. We fully agree with the reviewer that the magnitudes of direct solar radiative forcing of aerosol particles are affected by above, surrounding, and underlying clouds. The information of day-by-day variation of cloud fraction is valuable for estimation of solar radiation reaching the land surface. However, the cloud-induced impact on the radiation at surface is beyond the direct scope of this paper. "The focus of this study is to quantify of direct *ASRF* and *ASRFE* at the TOA, BOA, and in ATM under cloud-free sky conditions …" (see Paragraph 1 of the subsection "3.2 Radiative transfer simulation"). The cloud-free conditions were controlled by cloud screening and quality assurance procedures utilizing multi-angle observations of the sun-sky radiometer through the almucantar and principal plane scans in the entire sky before inversion (Smirnov et al., 2000; Holben et al., 2006; Li et al., 2015, 2018; Giles et al., 2019). The measurements of all sky view camera were also adopted as the ancillary evidences to assess cloud presence in this study. Inevitably, few small clouds out of the observation directions and super-thin clouds may escape from the cloud detection processing. They have impacts on the radiation reaching the surface more or less. However, to obtain the fraction of these clouds, the existing cloud detection methods should be significantly improved. We thank the reviewer for pushing us on this point and this issue will be considered in subsequent research.

References:

Giles, D. M., Sinyuk, A., Sorokin, M. S., Schafer J. S., Smirnov, A., Slutsker, I., Eck, T. F., Holben, B. N., Lewis, J. R., Campbell, J. R., Welton, E. J., Korkin, S. V., and Lyapustin, A. I.: Advancements in the Aerosol Robotic Network (AERONET) Version 3 database - Automated near-real-time quality control algorithm with improved cloud screening for Sun photometer aerosol optical depth (AOD) measurements, Atmospheric Measurement Techniques, 12, 169-209, doi:10.5194/amt-12-169-2019, 2019.

Holben, B. N., Eck, T. F., Slutsker, I., Smirnov, A., Sinyuk, A., Schafer, J., Giles, D., and Dubovik, O. : Aeronet's version 2.0 quality assurance criteria, Proceedings of SPIE - The International Society for Optical Engineering, 6408, doi: 10.1117/12.706524, 2006.

Li, Z., Li, D., Li, K., Xu, H., Cheng, X., Chen, C., Xie, Y., Li, L., Li, L., Li, W., Lv, Y., Qie, L., Zhang, Y., and Gu, X.: Sun/sky radiometer observation network with the extension of multi-wavelength polarization measurements, Journal of Remote Sensing, 19, 496-520, doi:10.11834/jrs.20144129, 2015.

Li, Z. Q., Xu, H., Li, K. T., Li, D. H., Xie, Y. S., Li, L., Zhang, Y., Gu, X. F., Zhao, W., Tian, Q. J., Deng, R. R., Su, X. L., Huang, B., Qiao, Y. L., Cui, W. Y., Hu, Y., Gong, C. L., Wang, Y. Q., Wang, X. F., Wang, J. P., Du, W. B., Pan, Z. Q., Li, Z. Z., and Bu, D.: Comprehensive Study of Optical, Physical, Chemical, and Radiative Properties of Total Columnar Atmospheric Aerosols over China: An Overview of Sun-Sky Radiometer Observation Network (SONET) Measurements, Bulletin of the American Meteorological Society, 99, 739-755, doi:10.1175/BAMS-D-17-0133.1, 2018.

Smirnov, A., Holben, B. N., Eck, T. F., Dubovik, O., and Slutsker, I.: Cloud-Screening and Quality Control Algorithms for the AERONET Database, Remote Sensing of Environment, 73(3), 337-349, doi:10.1016/S0034-4257(00)00109-7, 2000.

*3. Figure 11: The readers would like to know how the ASRF is derived from AERONET, instead of the performance of ASRF product. The details will shed light on the difference between ASRF from RT model and ASRF from AERONET.*

Reply: In response to this comment we have added more details about how the *ASRF* is derived from AERONET and the difference from the AERONET definition. Please find them in the first paragraph of subsection "4.3 Difference from AERONET products" in the revised manuscript:

**"For AERONET, broadband upward and downward irradiances in the SW ranges from 0.2 to 4.0 μm were calculated by radiative transfer model with retrieved aerosol properties as model inputs (http://aeronet.gsfc.nasa.gov). However, AERONET adopts different definition of *ASRF* that only taking the downward irradiance at the BOA and the upward irradiance at the TOA into consideration (García et al., 2012). The upward irradiances with and without aerosols in Eq. (2), along with the downward irradiances with and without aerosols in Eq. (1), are not taken into account. Omitting the downward irradiances will not make much difference in *ASRF* at the TOA. But for *ASRF* at the BOA, it is predictable that neglecting the upward irradiance will lead to obvious difference."**

---

## Author Response (AR1)

**COVER LETTER**

Manuscript ID: acp-2020-60

Dear Editor,

   We would like to thank you for giving us the opportunity to revise our manuscript, now titled "Aerosol solar radiative forcing near the Taklimakan Desert based on radiative transfer and regional meteorological simulations during the Dust Aerosol Observation-Kashi campaign". We appreciate the reviewers for thoughtful review of previous manuscript. They raised important issues and the comments are very helpful to improve the manuscript. We have carefully taken these comments into consideration and have revised our manuscript accordingly. The changes within the manuscript were highlighted using red text in the red-lined version. We are confident that the new version of the manuscript is significantly improved.

   We respond in detail to each of the reviewers' comments in the author response files. Please find the reviewers' comments repeated in blue italics, and followed by our responses in black and revisions in bold. We hope that the reviewers will find our responses to their comments satisfactory.

   Thanks for all the help!

Sincerely,

Li Li

lili3@radi.ac.cn

**To Anonymous Reviewer #1: (Comment ID:acp-2020-60-RC1)**

Dear Reviewer,

Thank you very much for carefully reviewing our manuscript. The constructive and thoughtful comments have helped us a lot to improve the paper. Following these suggestions, we have taken a lot of efforts to optimize the structure and to make the writing more concise and clear. Please find below the comments in blue italics and our responses in black and the changes in bold.

Responses to general comments:

*In this manuscript, the measurements obtained during the Dust Aerosol Observation-Kashi campaign were employed in radiative transfer model and the estimations were improved by considering the actual measured atmospheric profiles and diurnal variations of land surface albedo. Direct aerosol solar radiative forcing of dust aerosols was analyzed based on comprehensive parameters and numerical models. The effects of data assimilations on estimating the radiative forcing effects were also explored. However, the manuscript was poorly worded, thus making me confused. The manuscript needs to structure writing accurately to produce proper paragraphs with clear topics. Major revisions are necessary before the manuscript is finally accepted for publication.*

Reply: We thank the reviewer for the constructive criticisms that have helped us to improve our manuscript. We have worked hard on optimizing the structure of the manuscript. In the revised manuscript, we reorganized some sections:

1) subsection **"2.1 Experimental site and instrumentation" was divided into "2.1 Observation site" and "2.2 Instrumentation"**;

2) subsection **"2.2 Aerosol properties during the DAO-K campaign" was incorporated into subsection " 4.1 Aerosol solar radiative forcing and efficiency"**;

3) subsection **"3.3.3 Experimental setup" was rephrased as "3.3.3 Model setup"** to avoid confuse with the subsection of " Instrumentation";

4) section **"4 Results and Discussion" was changed to "4 Results of radiative transfer simulations"**;

5) subsection **"4.3 Comparisons and validation" was isolated from section 4 and changed to "5 Comparison with WRF-Chem simulations"** to avoid too much contents in section 4;

6) subsection **"4.3.1 Comparison between radiative transfer simulations and AERONET results" was renamed as "4.3 Difference from AERONET products"**, and was moved to follow the section 4.2 to state and discuss the results of radiative transfer model simulations together.

The new structure contains 6 parts: 1) Section "1 Introduction" begins with the introduction of background and significance, current status and problems, as well as research mentality and content of this manuscript. 2) Section "2 Dust Aerosol Observation-Kashi field campaign" gives a brief introduction of the DAO-K field campaign, and an overview of the multi-source observations and data. 3) Section "3 Estimation of aerosol solar radiative forcing" describes the methods to estimate *ASRF* by improving the inputs of atmospheric profiles and land surface albedo in RT simulation, and by employing data assimilation in the WRF-Chem simulation. 4) Section "4 Results of radiative transfer simulations" presents the results of *ASRF* simulated by RT model during the field campaign and for some specific cases. The influences of the atmosphere and surface conditions on the results are discussed. The difference from the corresponding AERONET operational products are also analyzed. 5) Section "5  Comparison with WRF-Chem simulations" gives direct comparison between the RT and WRF-Chem model simulations. Both the model simulations are evaluated based on the simultaneous irradiance measurements. 6) "Summary and conclusions" are given in Section 6. We believe that the new structure is more concise and understandable. We hope our revisions have satisfactorily addressed these issues.

Responses to specific comments:

*1. Section 1, this part should introduce the research background and significance, current status, concealed problems, as well as research mentality and content of this manuscript. Nevertheless, the introduction of this manuscript is inundated with accumulation of literatures rather than sublimation of these research results. The authors need survey more literatures in recent five years and then summarized them.*

Reply: Thank you for the frank comments and helpful suggestions. Following the suggestions, we have spent a lot of time to review and summarize related literatures. And on that basis we

restructured the "introduction" section and clarify the research background and significance, current status, concealed problems, as well as research mentality and content of this study. We have made a number of changes in these respects. Major revisions concentrated in the second and third paragraphs:

"**As one of the largest sandy deserts in the world, the Taklimakan Desert located in the Xinjiang Uygur Autonomous Region of China is a main source region of Asian dust (Huang et al., 2009), which influences not only surrounding areas such as the Tibetan Plateau (Liu et al., 2008; Chen et al., 2013; Yuan et al., 2019), but also wide regions in Eastern Asia (Mikami et al., 2006; Liu et al., 2011b; Yuan et al., 2019), even North America and Greenland through long-range transports across the Pacific Ocean (Bory et al., 2003; Chen et al., 2017; Liu et al., 2019). An accurate assessment of the Taklimakan aerosol solar radiative forcing (*ASRF*, defined as the difference of the net solar irradiance with and without aerosols presence) is important to evaluate regional and global climate changes. However, simulations by different models with different observation inputs varied widely in literatures. Huang et al. (2009) employed the Fu-Liou RT model to simulate the Taklimakan *ASRF* during the dust episodes in the summer of 2006, and reported that the dust particles result in average daily mean SW warming of 14 W m$^{-2}$ at the top of atmosphere (TOA), atmospheric warming of 79 W m$^{-2}$, and surface cooling of -65 W m$^{-2}$. Sun et al. (2012) adopted the RegCM4 simulations and reported both negative *ASRF* (i.e., cooling effects) of dust particles at the TOA and bottom of atmosphere (BOA) with the strongest values in spring during 2000~2009 period, reaching up to -4 W m$^{-2}$ and -25 W m$^{-2}$ in the Taklimakan Desert region, respectively. Li et al. (2018) also reported the negative multi-year average SW aerosol radiative forcing of -16 W m$^{-2}$ at the TOA and -18 W m$^{-2}$ at the BOA at the edge of Taklimakan Desert, Kashi station based on the SBDART simulations. The simulated results of dust aerosol radiative forcing have rarely been confirmed, especially in the Taklimakan Desert (Xia et al., 2009). Performances of various models sometimes were evaluated against the observations of aerosol optical depth (*AOD*), aerosol extinction profile, single scattering albedo (*SSA*), and particle size distribution (Zhao et al., 2010; Chen et al., 2014). Nevertheless, comparison of irradiance is indispensable to provide direct evidence for corroborating the *ASRF* simulated results.**"

"An intensive dust field campaign is essential for comprehensive investigating the optical, physical, chemical, and radiative properties of dust aerosol particles over Taklimakan Desert. As such, one of the goals of the Dust Aerosol Observation-Kashi (DAO-K) field campaign is to provide high quality dataset on aerosol in this region to obtain accurate assessment of the Taklimakan aerosol solar radiative forcing..."

Please find them in the section "1 Introduction" in the revised manuscript.

References:

[revised manuscript text omitted]

*2. Section 2 and 3, the authors seem to be drowned by abundant resources and avoid stringing them together to form a system. Some descriptions should be streamlined. The outline and structure of this manuscript should be reorganized.*

Reply: Thank you also for pointing out these issues on the structure and the descriptions in Sections 2 and 3. We have made a number of changes in these respects.

First of all, as mentioned in the responds to the general comments, we restructured section 2 and rephrased the title of subsection 3.3.3 to make the structure more clear and compact: 1) subsection **"2.2 Aerosol properties during the DAO-K campaign" was moved out from this section**; 2) subsection **"2.1 Experimental site and instrumentation" was divided into "2.1 Observation site" and "2.2 Instrumentation"**; 3) subsection **"3.3.3 Experimental setup" was rephrased as "3.3.3 Model setup"** to avoid confuse with the subsection of "Instrumentation".

Secondly, we **modified the subsection "2.1 Observation site"** to explain why the experimental site was selected in Kashi instead of other main source region of Asian dust (e.g., Gobi Desert) and the representativeness of the experimental period to study the dust radiative forcing effects. Fig. 1 was also edited to focus on the location of experimental site.

Finally, we **reorganized Table 1 in subsection "2.2 Instrumentation" to summarize the parameters and instruments in three groups (i.e., applications in radiative transfer simulation, WRF-Chem simulation, as well as evidences and validation)**. The first two groups of parameters work as model inputs in section 3 (i.e., inputs of radiative transfer simulation and WRF-Chem model simulation, respectively). **Correspondingly, the introductions of experimental apparatus and data in this subsection were sorted into three groups and were arranged in three paragraphs**: 1) measurements of main data of aerosol properties (including sun-sky radiometer, continuous particulate monitor, and ambient air quality continuous automated monitoring system); 2) measurements of ancillary parameters of surface albedo and the vertical structure (including sounding balloons, OMI/Aura, MODIS/Terra+Aqua, pyrheliometer and pyranometers); 3) measurements of ancillary evidences of dust and cloud layers (including LILAS lidar and all sky view camera). The descriptions of each group contained data processing, quality control, and applications in this study. **Some detailed descriptions about the calibration of the sun-sky radiometers were removed to make the main line more concise.** Please find them the new subsection "2.2 Instrumentation" in the revised manuscript.

In these ways we hope to have tightened the structure and optimized the descriptions of the Section 2 and 3.

*3. Section 2.1, this part should explain why the experimental site was selected in Kashi instead of the local aerosol properties, such as the representativeness or speciality in studying aerosol-related issues.*

Reply: In response to the above comment (specific comment 2) we divided subsection"2.1 Experimental site and instrumentation" into two subsections "2.1 Observation site" and "2.2 Instrumentation" to make this part expressed in much cleaner, more structured manner. Following this comment, we have revised the subsection on experimental site so that the specialitis of aerosol property and aerosol radiation effect at Kashi site are more carefully introduced:

**"In addition to the Kashi station near the Taklimakan Desert, SONET also maintains two dust aerosol observation stations (i.e., Zhangye and Minqin stations) in the Gobi Desert which is another important source of Asian dust. Although some studies reported that the dust generated in Taklimakan Desert exerts a less influence on long-range downstream regions due to the unique terrain and low-level background wind climatology compared to those in Gobi Desert (Chen et al., 2017; Liu et al., 2019), Taklimakan Desert is more representative to study the effects of dust aerosol solar radiative forcing on local region than the Gobi Desert because of its huge dust emission capability (Chen et al., 2017)."**

**"According to the SONET long-term measurements from 2013, the Kashi site is frequently affected by dust, where the multi-year average *AOD* is up to 0.56±0.18 at 500 nm; moreover, the Ångström exponent (*AE*, 440~870 nm) and fine-mode fraction (*FMF*, 500 nm) at Kashi are the lowest (with the multi-year average values of 0.54±0.27 and 0.40±0.14, respectively) among all 16 sites within SONET around China (Li et al., 2018). In contrast, the multiyear average *AOD*s (500 nm) at Zhangye (0.28±0.11) and Minqin (0.26±0.11) are only half of that at Kashi or less (Li et al., 2018). Meanwhile, their average values of *AE* and *FMF* are also greater than those at Kashi (Li et al., 2018). They all imply coarse particles are more dominant in the Taklimakan Desert in comparison with the Gobi Desert."**

Please find them in the subsection "2.1 Observation site" in the revised manuscript.

References:

Chen, S., Huang, J., Li, J., Jia, R., Jiang, N., Kang, L., Ma, X., and Xie, T.: Comparison of dust emissions, transport, and deposition between the Taklimakan Desert and Gobi Desert from 2007 to 2011. Science China Earth Sciences, 60(1), 1338-1355, doi:10.1007/s11430-016-9051-0, 2017.

Li, Z. Q., Xu, H., Li, K. T., Li, D. H., Xie, Y. S., Li, L., Zhang, Y., Gu, X. F., Zhao, W., Tian, Q. J., Deng, R. R., Su, X. L., Huang, B., Qiao, Y. L., Cui, W. Y., Hu, Y., Gong, C. L., Wang, Y. Q., Wang, X. F., Wang, J. P., Du, W. B., Pan, Z. Q., Li, Z. Z., and Bu, D.: Comprehensive Study of Optical, Physical, Chemical, and Radiative Properties of Total Columnar Atmospheric Aerosols over China: An Overview of Sun-Sky Radiometer Observation Network (SONET) Measurements, Bulletin of the American Meteorological Society, 99, 739-755, doi:10.1175/BAMS-D-17-0133.1, 2018.

Liu, L., Guo, J., Gong, H., Li, Z., Chen, W., Wu, R., Wang, L., Xu, H., Li, J., Chen, D., and Zhai, P.: Contrasting Influence of Gobi and Taklimakan Deserts on the Dust Aerosols in Western North America. Geophysical Research Letters, 46(15), 9064-9071, doi:10.1029/2019GL083508, 2019.

**4. Lines 85-86, 'the Ångström exponent (AE, 440~870 nm) and fine-mode fraction (FMF) at Kashi are the lowest among all sites in China' What is the scientific value of this sentence? And it needs strong literature to support.**

Reply: We changed this sentence into:

"…moreover, the Ångström exponent (*AE*, 440~870 nm) and fine-mode fraction (*FMF*, 500 nm) at Kashi are the lowest (with the multi-year average values of 0.54±0.27 and 0.40±0.14, respectively) among all 16 sites within SONET around China (Li et al., 2018)."

Please find it in the subsection "2.1 Observation site" in the revised manuscript. The scientific values of the multi-year average *AE* and *FMF* have been provided. The literature was also added.

References:

Li, Z. Q., Xu, H., Li, K. T., Li, D. H., Xie, Y. S., Li, L., Zhang, Y., Gu, X. F., Zhao, W., Tian, Q. J., Deng, R. R., Su, X. L., Huang, B., Qiao, Y. L., Cui, W. Y., Hu, Y., Gong, C. L., Wang, Y. Q., Wang, X. F., Wang, J. P., Du, W. B., Pan, Z. Q., Li, Z. Z., and Bu, D.: Comprehensive Study of Optical, Physical, Chemical, and Radiative Properties of Total Columnar Atmospheric Aerosols over China: An Overview of Sun-Sky Radiometer Observation Network (SONET) Measurements, Bulletin of the American Meteorological Society, 99, 739-755, doi:10.1175/BAMS-D-17-0133.1, 2018.

**5. Section 2.2, 'aerosol properties during the DAO-K campaign' is part of the 'Results', so I suggest moving it to Section 4.**

Reply: We agree with this point and have **moved the contents "aerosol properties during the DAO-K campaign" to section "4 Results of radiative transfer simulations"**. We realized that putting this part in section "2 Dust Aerosol Observation-Kashi field campaign" made the structure less understandable after the reviewer pointed it out. The reason being that in the previous manuscript these aerosol properties were input parameters for model simulations then were presented before the section "3 Estimation of aerosol solar radiation forcing". Following this

suggestion, we decided to include this part in the section of results (i.e., "4 Results of radiative transfer simulations"). We also reduced the descriptions on aerosol properties during the DAO-K campaign and focused on the aerosol properties relating to solar radiative forcing and efficiency. Please find them in the first paragraph in the subsection "4.1 Aerosol solar radiative forcing and efficiency".

*6. The structure of the manuscript makes me feel that some parts are more or less irrelevant to the title 'Solar radiative forcing of aerosol particles near the Taklimakan desert during the Dust Aerosol Observation-Kashi campaign in Spring 2019'. Too much attention was spent on Section 4.3.*

Reply: The improvements of dust radiative forcing estimation and the evaluation of the model results are the main points of this manuscript. We recognize that such improvement of solar radiative forcing estimation and the comprehensive evaluations of model results in the manuscript may provide meanings for dust radiative forcing research in different regions and also can be extended to other kind of aerosol particles. Besides structuring contents into a more understandable format (see the reply to the general comments), we changed the title into "**Aerosol solar radiative forcing near the Taklimakan Desert based on radiative transfer and regional meteorological simulations during the Dust Aerosol Observation-Kashi campaign**" to make the topic more clearly and completely to be expressed.

To avoid too much contents in section 4 and to highlight the comparison and evaluation of the radiative transfer and WRF-Chem simulations in the manuscript, **subsection "4.3 Comparisons and validation" was isolated from section 4 and changed to "5 Comparison with WRF-Chem simulations".** The old subsection "4.3.1 Comparison between radiative transfer simulations and AERONET results" was renamed as "4.3 Difference from AERONET products", and was moved to follow the section 4.2 to state and discuss the results of radiative transfer simulations together.

Responses to technical comments:

*1. Line 17, 'are improved by' should be changed to 'were improved by'.*

Reply: **Changed as suggested**. Please see line 17 in the revised manuscript.

*2. Line 40, 'it is a challenging' should be changed to 'it is challenging' or 'it is a challenge'.*

Reply: It has been **changed to "it is still challenging"**. Please see line 39 in the revised manuscript.

*3. Line 41, add 'the' before 'high surface albedo over desert'.*

Reply: **Added as suggested.** Please see line 40 in the revised manuscript.

*4. Line 42, I suggested replacing the sentence 'Numerous efforts have investigated…' with 'Numerous efforts have been undertaken to investigate…'.*

Reply: **Replaced as suggested.** Please see lines 34-35 in the revised manuscript.

*5. Line 52, 'have relatively small inter-annual variation' should be changed to 'had relatively small inter-annual variation'.*

Reply: **Changed as suggested.** Please see lines 101-102 in the revised manuscript.-

*6. Line 53, 'According to WRF-Chem simulations' should be changed to 'According to the WRF-Chem simulations'.*

Reply: **The sentence has been removed in the revised manuscript.**

*7. Line 85, the comma before 'moreover' should be changed to semicolon.*

Reply: **Changed as suggested.** Please see line 94 in the revised manuscript.

*8. Line 239, 'includes' should be changed to 'included'.*

Reply: **Changed as suggested.** Please see line 249 in the revised manuscript.

*9. Line 315, 'Globally' should be changed to 'Generally'.*

Reply: **Changed as suggested.** Please see line 345 in the revised manuscript.

*10. Line 433, 'for it will damage the surface-layer particulate results' should be changed to 'for that it will damage the surface-layer particulate results'.*

Reply: **Changed as suggested.** Please see lines 460-461 in the revised manuscript.

*11. I suggest deleting some acronyms, especially the phrases only appear once. Too many acronyms make the article chaotic.*

Reply: Some less common acronyms, like **" TD" (Taklimakan Desert), " LOA" (Laboratoire d'Optique Atmosphérique), " CNEMS" (China National Environmental Monitoring Center), "BEC" (background error covariance), were deleted** in the revised manuscript.

**To Anonymous Reviewer #2: (Comment ID:acp-2020-60-RC2)**

Dear Reviewer,

Thank you very much for your detailed and supportive comments. You raised important issues in the comments. They can help us to improve the manuscript to a better scientific level. We have carefully taken these comments into consideration and have revised our manuscript accordingly. Please find below the comments in blue italics and our responses in black and the changes in bold.

Responses to major comments:

*1. P2L35-38: This sentence is confusing to me. If my understanding is right, it is meant to express the dust originated from Taklimakan Desert (TD) exerts influences the air quality and climate over the downstream regions via long-range transport. Therefore, please try to be specific instead of using general words. However, some key references are missing, since both observations (Liu et al., 2019, doi:10.1029/2019GL083508.) and model simulations (Chen et al., 2017, doi: 10.1007/s11430-016-9051-0) suggested that the dusts generated in TD have LESS impacts on downstream regions due to the unique terrain and low-level background wind climatology, compared with those from other deserts in northwestern China.*

Reply: We thank the reviewer for pointing out these issues around influences of dust aerosol particles originated from Taklimakan Desert. In response to the comments, we have rephrased this sentence in a more specific way. The suggested references have been added. Please find them in the second paragraph of section "1 Introduction" in the revised manuscript:

"**As one of the largest sandy deserts in the world, the Taklimakan Desert located in the Xinjiang Uygur Autonomous Region of China is a main source region of Asian dust (Huang et al., 2009), which influences not only surrounding areas such as the Tibetan Plateau (Liu et al., 2008; Chen et al., 2013; Yuan et al., 2019), but also wide regions in Eastern Asia (Mikami et al., 2006; Liu et al., 2011b; Yuan et al., 2019), even North America and Greenland through long-range transports across the Pacific Ocean (Bory et al., 2003; Chen et al., 2017; Liu et al., 2019).**"

The issue of less impacts of Taklimakan dust particles on downstream regions due to the unique terrain and low-level background wind climatology, and the reason that the experimental site was

selected near Taklimakan desert instead of the Gobi desert were also referred and discussed. Please find them in the subsection "2.1 Observation site" in the revised manuscript:

[revised manuscript text omitted]

*2. Figures 3, 5, 10, 12: The X-axis can be considered to be revised (more minor ticks and labels are needed to be given), given the ASRF and ASRFE are only able to be estimated during daytime without clouds. Another important issue is the cloud-induced impact on the radiation reaching the surface. The authors are better to analyze the day-by-day variation of cloud (fraction) over the study sites of Kashi, which is concurrent with the ground-based aerosol remote sensing and radiation observations. I believe this will provide more insights into the community of aerosol radiative forcing.*

Reply: **More minor ticks and labels of the x-axes have been added in Figs. 4 (old Fig. 3), 5, 10**. For Fig. 12, in order to compare the WRF-Chem simulations and observations point by point, we interpolated the observations of $PM_{2.5}$, $PM_{10}$, and $AOD$ at 675 nm to the corresponding assimilation time of 0:00, 6:00, 12:00 and 18:00 UTC of each day. Please also see the statement "The second one-month simulation was assimilated the observations of $PM_{2.5}$, $PM_{10}$ and $AOD$ with GSI at 0:00, 6:00, 12:00 and 18:00 UTC with the assimilation window of ±3 h centered at the analysis time." in the first paragraph of subsection "3.3.3 Model setup". So, the ticks and labels were given with daily intervals in Fig. 12. For each day, there are four sets of $PM_{2.5}$, $PM_{10}$ in Figs. 12a and c. But normally less than four sets of $AOD$ can be obtained owing to cloud screening or other quality control operation (see Fig. 12e). **To clarify this, the caption of Fig. 12 has been changed into:** "**Comparisons of the surface-layer $PM_{2.5}$ (a, b), $PM_{10}$ (c, d) concentrations and $AOD$ at 675 nm (e, f) among the observations, the WRF-Chem simulations with and without data assimilations (DA) in April 2019. The observations have been interpolated to 0:00, 6:00, 12:00, 18:00 UTC of each day.**"

We thank the comment on cloud-induced impacts which raises an important issue. We fully agree with the reviewer that the magnitudes of direct solar radiative forcing of aerosol particles are affected by above, surrounding, and underlying clouds. The information of day-by-day variation of cloud fraction is valuable for estimation of solar radiation reaching the land surface. However, the cloud-induced impact on the radiation at surface is beyond the direct scope of this paper. "The focus of this study is to quantify of direct *ASRF* and *ASRFE* at the TOA, BOA, and in ATM under cloud-free sky conditions …" (see Paragraph 1 of the subsection "3.2 Radiative transfer simulation"). The cloud-free conditions were controlled by cloud screening and quality assurance procedures utilizing multi-angle observations of the sun-sky radiometer through the almucantar and principal plane scans in the entire sky before inversion (Smirnov et al., 2000; Holben et al., 2006; Li et al., 2015, 2018; Giles et al., 2019). The measurements of all sky view camera were also adopted as the ancillary evidences to assess cloud presence in this study. Inevitably, few small clouds out of the observation directions and super-thin clouds may escape from the cloud detection processing. They have impacts on the radiation reaching the surface more or less. However, to obtain the fraction of these clouds, the existing cloud detection methods should be significantly improved. We thank the reviewer for pushing us on this point and this issue will be considered in subsequent research.

criteria (Li et al., 2018).  Observations from the CE318 #1141  also joined in the AERONET dataset. The consistency of the products following the AERONET and SONET retrieval frameworks has been validated by Li et al. (2018).  The multi-wavelength properties of *AOD*, *SSA*, *AE*, and *g* were applied in radiative transfer model simulations. In addition to sun-sky radiometers, a METONE BAM-1020 Continuous Particulate Monitor was also deployed to measure $PM_{2.5}$ mass concentration (mg m$^{-3}$) (Fig. 2a). The hourly $PM_{10}$ mass concentration (mg m$^{-3}$) were collected from the routine measurements of ambient air quality continuous automated monitoring system in Kashi operated by China National Environmental Monitoring Center. The aerosol parameters including *AOD*, $PM_{2.5}$ and $PM_{10}$ mass concentrations were assimilated in WRF-Chem model simulation in this study.

Aerosol radiative effects also depend on the surface albedo and the vertical structure of atmosphere. During the campaign, atmospheric profiles, including the vertical distributions of the atmospheric pressure, temperature, and relative humidity, were collected from sounding balloon measurements operated by Kashi regional meteorological bureau. Data quality was controlled following the operational specifications for conventional upper-air meteorological observations (China Meteorological Administration, 2010). The sounding balloons incorporate radiosondes were regularly launched twice a day around 0:00 and 12:00 UTC at Kashi weather station (39.46°N, 75.98°E, 1291 m above mean sea level). Normally there were more than 60 layers were specified from land surface to over 35 km.  In addition to pressure, temperature, and relative humidity profiles, ozone profiles obtained by the Ozone Monitoring Instrument (OMI)/Aura satellite (Bhartia et al., 1996) were also adopted as the RT model inputs. The satellite observations of Moderate resolution imaging spectroradiometer (MODIS)/Terra+Aqua were employed to collect the surface information during the DAO-K campaign.  The MODIS products of shortwave bidirectional reflectance distribution function (BRDF) parameters, black-sky albedo (BSA), and white-sky albedo (WSA) were adopted to derive the surface albedo during the daytime (Schaaf and Wang, 2015). A solar radiation monitoring station, equipped with an EKO MS-57 pyrheliometer and two MS-80 pyranometers, was used for measuring the direct, diffuse, and total solar irradiances (W m$^{-2}$) in the range of 280~3000 nm (Fig. 2a). The pyrheliometer and pyranometers have been calibrated before the campaign with uncertainties of 0.55% and 0.66%, respectively. They satisfy the requirements of class A under the ISO 9060:2018 with fast response time <0.2 s and <0.4 s, separately, which make them have excellent performances in understanding of the dynamics of solar irradiances in the atmosphere. The high-quality dataset of direct, diffuse, and total downward irradiances was applied in evaluation and validation of the RT and WRF-Chem simulations. The fraction of diffuse skylight radiation deduced from the diffuse and total irradiances also gave a key weighting index to modulate the diurnal-changes of the surface albedo.

Moreover, some other instruments provided independent evidences of the existences of dust and cloud layers in the atmosphere. Multiwavelength Mie-Raman polarization lidar (LILAS) developed by the Laboratoire d'Optique Atmosphérique,

185  (LOA) of University of Lille Université de Lille 1(Fig. 2b), was equipped with three elastic wavelengths (all linearly polarized) at 355, 532, 1064 nm and three Raman wavelengths at 387, 530, 408 nm, from which the vertical distribution of multiple optical and physical properties of dust aerosol particles can be obtained (Veselovskii et al., 2016, 2018; Hu et al., 2019). The backscattering coefficient profile at 355 nm wavelength was applied in this study to distinguish the two-layer structure of dust.

190   The YNT all sky view camera ASC200  equipped with two wide-dynamic full-sky visible and infrared imagers,  recorded dynamic states of clouds  in the whole sky during day and night with 10 min (or less than 10 min) resolution.

195   An overview of the instruments and corresponding parameters employed in the study is listed in Table 1. Considering different durations of various measurements,  we calculated and discussed the *ASRF* from 2 to 25 April 2019, when simultaneous measurements are available.

[Figure]

200  **Figure 2: Setup of experimental apparatus during the DAO-K field campaign (a) on the roof, (b) in door.**

**Table 1: Overview of the Parameters and instruments employed in the radiative transfer and WRF-Chem model simulations and validation. **

[revised manuscript text omitted]

---

## Author Response (AR3)

**To Anonymous Referee:**

Dear Reviewer,

Thank you very much for carefully reviewing the revised manuscript. Please find below the comments in blue italics and our responses in black and the changes in bold.

Responses to specific comments:

*1. To be honest, i do not like the reply to my comment #8, most of which are flawed: For instance:*

*a) Data quality issue of radiosonde measurements: "Operational specifications for conventional upper-air meteorological observations" issued by CMA is way too outdated, the authors are strongly recommended to refer to Zhang et al., 2018 (doi:10.1175/JCLI-D-17-0231.1) and Guo et al. 2019 (doi:10.1029/2019GL082666) and the references therein for more details.*

*b) There is grammatical errors in "The sounding balloons incorporate radiosondes were regularly..", which can be considered revised to "The sounding balloons were operationally.."*

*c) "60 layers" is misleading. Actually, there is typically about 3000 layers given the sampling frequency of 1 second for the balloons (c.f. 5 - 8 m resolution in the vertical)*

*d) "35 km": for most of the soundings, it is hard for them to reach such altitudes. the authors can revise it to "Normally there were 3000 measurements, more or less, recorded during the flight path of balloon (Guo et al., 2016, doi:10.5194/acp-16-13309-2016)."*

Reply: We thank the reviewer for the constructive criticisms and providing references.

a) We have carefully review the suggested references and changed the statements about the data quality of sounding balloon measurements into:

"**The data quality was controlled following the operational specifications for conventional upper-air meteorological observations (China Meteorological Administration, 2010). The accuracy of the temperature profile in the troposphere is within ±0.1 K (Zhang et al., 2018; Guo et al, 2019).**"

Please find them in lines 140-142 in the revised manuscript. The suggested references have been added.

b) Following the suggestion, we changed this sentence into:

"**The sounding balloons were operationally launched twice a day around 0:00 UTC and 12:00 UTC at Kashi weather station (39.46°N, 75.98°E, 1291 m above mean sea level).**"

Please find it in lines 137-139 in the revised manuscript.

c)   Here we were trying to say that more than 60 layers were specified in the SBDART simulations in this study. This sentence might be misleading. We fully agree with this comment and rephrased this sentence to:

"**Normally, about 3000 individual measurements are recorded during one balloon flight, which corresponds to a sampling frequency of 1 second (Guo et al., 2016; Chen et al., 2019).**"

Please find it in lines 139-140 in the revised manuscript.

d)   We have carefully review the suggested and related references. The statement of "from land surface to over 35 km" has been removed. This sentence was rephrased to:

"**Normally, about 3000 individual measurements are recorded during one balloon flight, which corresponds to a sampling frequency of 1 second (Guo et al., 2016; Chen et al., 2019).**"

Please also see the reply to the above comment, The suggested and related references have been added.

Reply: **Changed as suggested.** Please see line 41 in the revised manuscript.

[revised manuscript text omitted]